# MASC: Boosting Autoregressive Image Generation with a Manifold-Aligned Semantic Clustering

## Abstract

Autoregressive (AR) models have shown great promise in image generation, yet they face a fundamental inefficiency stemming from their core component: a vast, unstructured vocabulary of visual tokens. This conventional approach treats tokens as a flat vocabulary, disregarding the intrinsic structure of the token embedding space where proximity often correlates with semantic similarity. This oversight results in a highly complex prediction task, which hinders training efficiency and limits final generation quality. To resolve this, we propose **M**anifold-**A**ligned **S**emantic **C**lustering (MASC), a principled framework that constructs a hierarchical semantic tree directly from the codebook's intrinsic structure. MASC employs a novel geometry-aware distance metric and a density-driven agglomerative construction to model the underlying manifold of the token embeddings. By transforming the flat, high-dimensional prediction task into a structured, hierarchical one, MASC introduces a beneficial inductive bias that significantly simplifies the learning problem for the AR model. MASC is designed as a plug-and-play module, and our extensive experiments validate its effectiveness: it accelerates training by up to 71% and significantly improves generation quality, reducing the FID of LlamaGen-XL from 2.87 to 2.58. MASC elevates existing AR frameworks to be highly competitive with state-of-the-art methods, establishing that structuring the prediction space is as crucial as architectural innovation for scalable generative modeling. Our code is open-sourced via `https://anonymous.4open.science/r/anonymous_MASC-F3D2/`.

## 1 Introduction

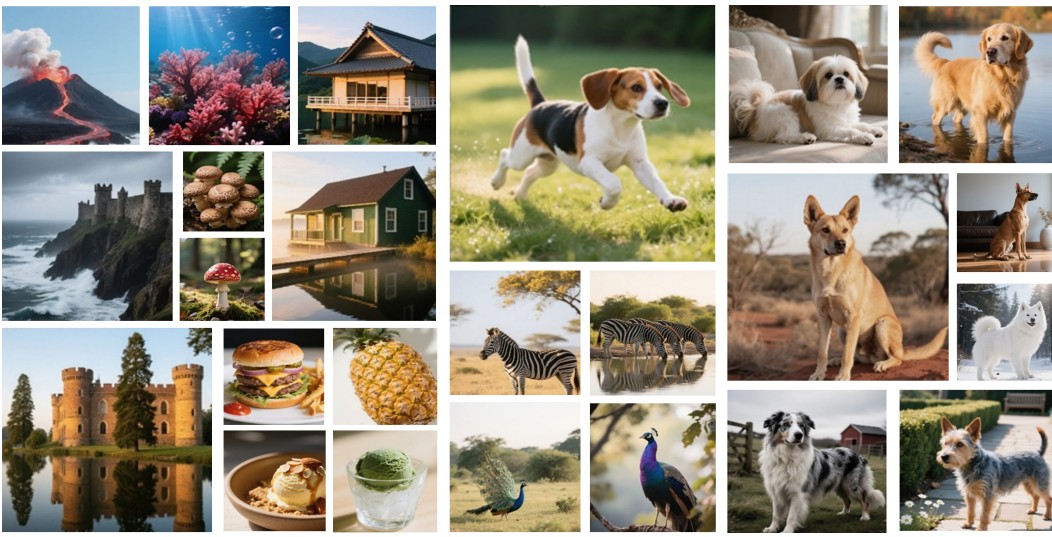

Figure 1: MASC demonstrates strong capabilities in high-fidelity and diverse image generation.

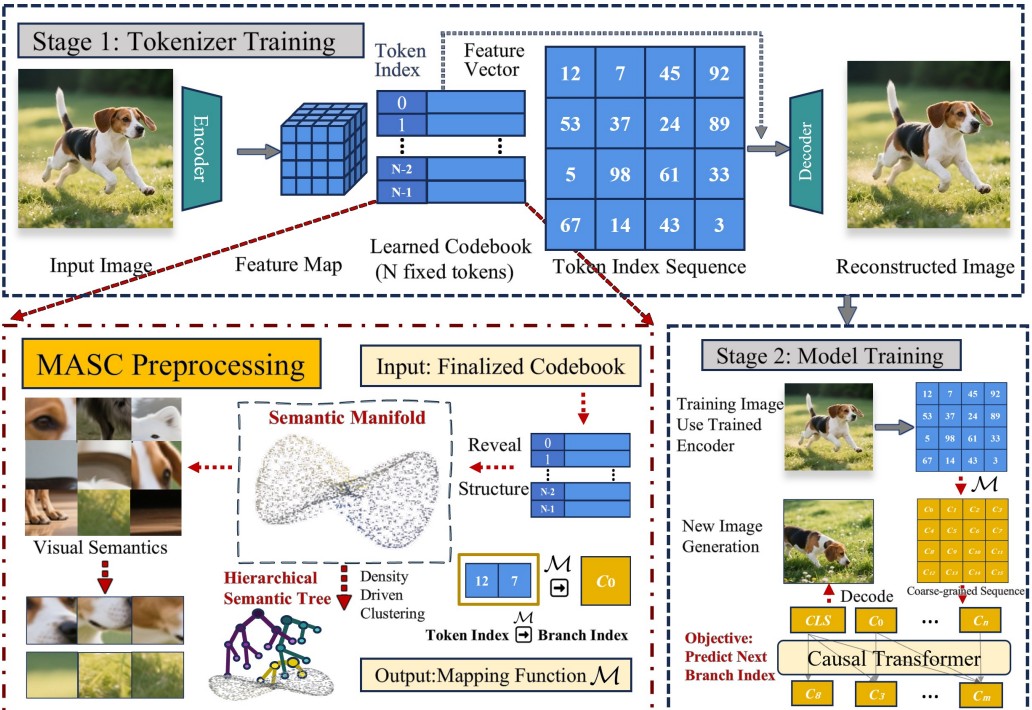

Figure 2: Overview of the MASC-integrated autoregressive generation pipeline, and the bottom row showcases high-fidelity images generated by MASC. **Stage 1:** A standard image tokenizer is trained, yielding a finalized codebook of visual tokens. **MASC Preprocessing:** Our proposed MASC framework takes this codebook and constructs a hierarchical semantic tree, producing a mapping function ($\mathcal{M}$) from fine-grained tokens to coarse-grained semantic branches. **Stage 2:** The autoregressive Transformer is then trained on these simplified branch indices.

Autoregressive (AR) image generation (Esser et al., 2021; Sun et al., 2024; Tian et al., 2024; Zhou et al., 2024; Fan et al., 2024; Li et al., 2024) has developed rapidly, demonstrating impressive scalability analogous to Large Language Models (LLMs) (Brown et al., 2020; Touvron et al., 2023) and achieving a level of fidelity that rivals leading diffusion-based methods (Peebles & Xie, 2023; Dhariwal & Nichol, 2021; Hatamizadeh et al., 2024). Among the various approaches, those utilizing discrete tokenizers have garnered significant attention not only for their structural alignment with the well-established LLM paradigm but also for their training efficiency and straightforward optimization. As illustrated in Figure 2, the standard framework for these models consists of two distinct stages: an image tokenizer and an autoregressive causal Transformer. The tokenizer first converts a continuous image into a sequence of discrete integer indices via a learned codebook. Subsequently, the Transformer is trained on these index sequences to causally predict the next token.

The elegance of this paradigm, however, conceals a fundamental drawback stemming from the quantization process: a critical loss of structural information. The AR model is trained on a flat vocabulary of discrete indices, which implicitly treats semantically similar tokens (e.g., different shades of blue sky) as unrelated entries in a vast categorical space. This process largely disregards the rich geometric relationships embedded within the codebook's high-dimensional vector space. These token embeddings are not randomly scattered; rather, they lie on or near a lower-dimensional semantic manifold (Huh et al., 2023; Van Den Oord et al., 2017), where geometric proximity corresponds to semantic similarity. By ignoring this intrinsic structure, the AR model is forced to tackle a massive $N$-way classification problem (where $N$ can be 16,384 or more) at each step. Lacking any explicit prior about the relationships between tokens, the model must expend significant capacity and data to implicitly re-learn these structures from scratch, which imposes a tremendous learning burden, leading to sample inefficiency and slow convergence (Ranzato et al., 2015).

Recognizing this challenge, a recent line of work has sought to re-introduce this lost structural information by providing the AR model with a codebook prior (Hu et al., 2025; Guo et al., 2025). The common approach is to use k-means clustering to group the token embeddings. However, this

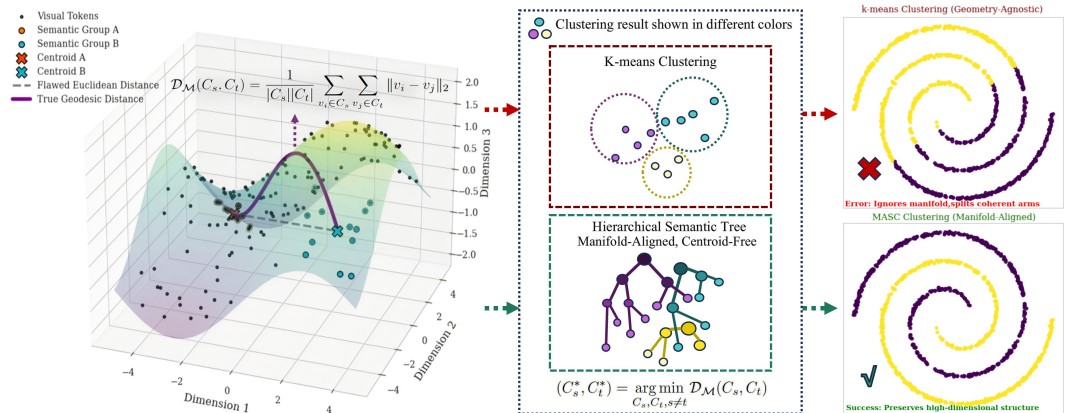

Figure 3: Conceptual illustration of MASC versus k-means. **Left:** Visual tokens reside on a semantic manifold where geodesic distance is a better similarity measure than Euclidean distance (dashed line). **Right:** Geometry-agnostic k-means produces incoherent clusters, whereas MASC's manifold-aligned approach correctly captures the intrinsic data structure and preserves semantic integrity.

strategy is fundamentally limited because k-means is ill-suited to the intrinsic structure of the semantic manifold. Standard Euclidean distance, which k-means relies on, is a poor proxy for true semantic distance on a curved manifold. Cluster centroids, calculated as simple arithmetic means, can fall off the manifold entirely, making them poor representatives of their clusters (Beyer et al., 1999; Huh et al., 2023). Furthermore, the distribution of tokens on this manifold is highly non-uniform, reflecting the statistics of the visual world (Van Den Oord et al., 2017). Standard k-means, being insensitive to density, often produces semantically incoherent clusters, yielding a noisy and counterproductive prior. This reveals that while identifying the problem of unstructured vocabularies is a step forward, solving it requires a more principled approach than naive clustering.

In this work, we introduce **M**anifold-**A**ligned **S**emantic **C**lustering (MASC), a framework designed to construct a principled, structure-aware prior that serves as a powerful inductive bias for the AR model. MASC directly confronts the aforementioned challenges with two key innovations: (1) a robust, manifold-aligned similarity metric that is centroid-free, and (2) a density-driven, agglomerative construction process that respects the non-uniform token distribution. The result is a hierarchical semantic tree that transforms the difficult, high-dimensional flat prediction problem into a simpler, structured one, as shown in Figure 3. This unlocks significant gains in both training efficiency and generation quality. Our main contributions are summarized as follows:

- We identify and analyze the problem of structural information loss in discrete AR models, where treating tokens as a flat, unstructured vocabulary complicates the prediction task.

- We propose **MASC**, a framework that constructs a structure-aware inductive bias by faithfully modeling the codebook's underlying manifold. Its manifold-aligned similarity metric and density-driven construction offer a principled alternative to naive, geometry-agnostic clustering methods like k-means.

- We provide extensive empirical validation demonstrating that MASC is a versatile plug-and-play module that accelerates training by up to **57%**, improves generation quality (e.g., reducing the FID of LlamaGen-XL from 2.87 to **2.58**), and offers a practical solution to a core challenge in the field.

## 2 RELATED WORK

**Autoregressive Image Generation.** Inspired by Large Language Models (LLMs) (Brown et al., 2020; Touvron et al., 2023), the field of autoregressive (AR) image generation has rapidly matured. Its core paradigm uses a tokenizer (Esser et al., 2021; Yu et al., 2021) to convert images into discrete sequences for a Transformer model, an approach that has proven highly scalable and achieved quality competitive with leading diffusion models (Dhariwal & Nichol, 2021; Peebles & Xie, 2023). A wave of architectural innovations has advanced the field, from adapting LLM designs (Sun et al., 2024) and exploring diverse prediction schemes (Tian et al., 2024; Zhou et al., 2024; Han et al., 2024; Fan et al.,

2024), to developing new decoding, refinement, and alignment strategies (Zhang et al., 2025; Cheng et al., 2025; Wu et al., 2025). Despite these significant advancements, a unifying challenge persists: discrete AR models perform prediction over a **large, flat, and unstructured vocabulary**. This focus on architecture has largely sidestepped the intrinsic inefficiency of the prediction task itself, which motivates our work to explicitly structure the prediction space.

**Codebook Priors and Manipulation.** Recognizing the limitations of a flat vocabulary, pioneering methods have sought to extract a codebook prior to simplify the AR model's training. The predominant approach has been to apply k-means clustering to the token embeddings (Hu et al., 2025; Guo et al., 2025). However, the reliance on a naive, geometry-agnostic algorithm like k-means is a fundamental limitation. The visual token space is better modeled as a non-uniform semantic manifold (Van Den Oord et al., 2017; Huh et al., 2023), where k-means' core assumptions about Euclidean space and meaningful centroids are violated. This mismatch often results in semantically incoherent clusters and a noisy prior (Beyer et al., 1999). Other works on codebook manipulation focus on improving the tokenizer's representational quality (Li et al., 2023; Zheng & Vedaldi, 2023) rather than providing a structural prior to ease the AR model's prediction task. This leaves a clear gap for a principled, manifold-aligned prior extraction framework.

**Manifold Learning and Hierarchical Clustering.** Our approach is grounded in two established principles. First, Manifold learning posits that token embeddings lie on a low-dimensional manifold (Tenenbaum et al., 2000; Belkin & Niyogi, 2003), where standard Euclidean distance is an unreliable similarity metric, thus necessitating geometry-aware methods (Beyer et al., 1999; Angiulli, 2018; Chen & Li, 2024). Second, hierarchical clustering (Lukasová, 1979; Ward Jr, 1963) is naturally suited for the non-uniform density of such data. MASC operationalizes a synthesis of these principles, employing a geometry-aware metric within a density-driven, hierarchical algorithm. This marks a shift from heuristic clustering to a principled, manifold-aligned construction of the prediction space.

## 3 METHODOLOGY

### 3.1 PRELIMINARIES: THE HIGH-DIMENSIONAL PREDICTION CHALLENGE

Discrete token-based autoregressive image generation frameworks (Esser et al., 2021; Sun et al., 2024; Tian et al., 2024) operate via a two-stage process. First, an image tokenizer maps a continuous image $x \in \mathbb{R}^{H \times W \times 3}$ into a sequence of discrete integer indices. An encoder $E$ produces a feature map, and each feature vector $z^{(i,j)}$ is quantized to its nearest token in a learnable codebook $\mathcal{Z} = \{v_1, \ldots, v_N\} \subset \mathbb{R}^d$ via a nearest-neighbor lookup:

$$q^{(i,j)} = \underset{k \in \{1,\ldots,N\}}{\arg\min} \|z^{(i,j)} - v_k\|_2, \tag{1}$$

resulting in a grid of token indices $z^q \in \{1, \ldots, N\}^{h \times w}$, which is flattened into a 1D sequence of length $L = h \times w$.

In the second stage, an autoregressive model $\mathcal{G}$, typically a Transformer, is trained to maximize the likelihood of this token sequence, predicting the next token $z_t^q$ based on the preceding context $z_{<t}^q$:

$$p(z^q) = \prod_{t=1}^{L} p(z_t^q | z_{<t}^q; \mathcal{G}). \tag{2}$$

This is practically realized by an output layer producing a probability distribution over all $N$ tokens. The core difficulty, which we term the high-dimensional prediction challenge, stems from this process. The model is forced to perform a massive $N$-way classification at every step, treating the vocabulary as a flat, unstructured set of categories. This quantization step discards the rich geometric structure of the codebook, where semantic relationships are encoded as distances. Without this structural information, the model must expend significant capacity learning the semantic landscape from scratch. A codebook prior, therefore, aims to re-introduce this latent structure, providing an inductive bias to make the prediction task more tractable and efficient.

### 3.2 MASC: CONSTRUCTING A STRUCTURE-AWARE PRIOR

To resolve the high-dimensional prediction challenge, we introduce **M**anifold-**A**ligned **S**emantic **C**lustering (MASC), a framework designed to construct a structure-aware prior from the codebook.

MASC builds a hierarchical semantic tree that models the intrinsic geometric and statistical properties of the token embeddings, providing a powerful and meaningful inductive bias for the AR model. The construction process is composed of two core components: a principled similarity metric and a density-driven clustering algorithm.

### 3.2.1 A Principled Distance for a Semantic Manifold

A principled prior must be built upon an accurate measure of semantic similarity. As previously discussed, the token embeddings $\{v_i\}$ lie on or near a complex semantic manifold embedded within a high-dimensional space. A naive approach like k-means, which relies on Euclidean distance to cluster centroids, is fundamentally flawed in this context for two primary reasons. First, the arithmetic mean used to compute a centroid is a point in the ambient Euclidean space, which may not lie on the manifold itself. Using such an off-manifold point as a representative for a group of on-manifold tokens is geometrically unsound and can lead to inaccurate cluster assignments (Huh et al., 2023). Second, even if the centroid were valid, standard Euclidean distance is a poor proxy for true semantic similarity in a high-dimensional, curved space, which is more accurately described by the path length along the manifold's surface (i.e., the geodesic distance) (Beyer et al., 1999; Tenenbaum et al., 2000).

While computing the true geodesic distance is computationally intractable for an implicitly defined manifold, we can instead devise a practical metric that respects the manifold's properties. Instead of relying on a single, potentially invalid centroid, we employ a robust, **instance-based average distance** to measure the dissimilarity between two clusters, $C_s$ and $C_t$. This metric aggregates the pairwise Euclidean distances between all tokens across the two clusters:

$$\mathcal{D}(C_s, C_t) = \frac{1}{|C_s||C_t|} \sum_{v_i \in C_s} \sum_{v_j \in C_t} \|v_i - v_j\|_2. \tag{3}$$

This metric is inherently manifold-aligned because it is centroid-free, relying exclusively on the locations of the actual tokens which are guaranteed to lie on the manifold. By averaging over all pairs, it provides a robust measure of inter-cluster separation that implicitly respects the local geometry and is less sensitive to outliers, serving as a sound basis for constructing a meaningful semantic hierarchy.

### 3.2.2 Density-Driven Hierarchical Construction

Equipped with our similarity metric $\mathcal{D}$, we construct the semantic tree using a bottom-up, agglomerative hierarchical strategy (Lukasová, 1979; Ward Jr, 1963). This choice is crucial as the algorithm is deterministic and inherently respects the non-uniform density distribution of tokens on the manifold. The construction process proceeds as follows:

**Initialization**: The process begins with $N$ initial clusters, where each cluster $C_j^{(0)}$ contains a single token from the codebook: $C_j^{(0)} = \{v_j\}$ for $j \in \{1, \ldots, N\}$. These form the leaf nodes of our tree.

**Iterative Merging**: The algorithm then performs $N - 1$ merging iterations. In each iteration $i$, it identifies the pair of distinct clusters $(C_s^*, C_t^*)$ from the current set of clusters $\mathbb{C}^{(i-1)}$ that are most similar to each other, and merges them to form a new parent cluster. The merging criterion is formally expressed as:

$$(C_s^*, C_t^*) = \underset{C_s, C_t \in \mathbb{C}^{(i-1)}, s \neq t}{\arg\min} \mathcal{D}(C_s, C_t). \tag{4}$$

The new set of clusters for the next iteration, $\mathbb{C}^{(i)}$, is then formed by replacing $C_s^*$ and $C_t^*$ with their union, $C_s^* \cup C_t^*$.

**Termination**: This merging process is repeated until only one cluster remains—the root of the MASC tree, which encompasses all $N$ original tokens. The complete sequence of merges defines the entire binary tree structure.

The critical insight is that this bottom-up construction is implicitly **density-driven**. High-density regions on the semantic manifold correspond to areas where many tokens with similar visual semantics are tightly packed. Consequently, the pairwise distances $\mathcal{D}$ between tokens or clusters within these regions will be small. Our algorithm, by always merging the closest pair according to Eq. 4, naturally prioritizes forming connections within these dense, semantically coherent regions first. This property avoids the critical flaw of methods like k-means, which can carelessly partition dense regions, and guarantees that the most fundamental semantic similarities are faithfully captured in the hierarchy.

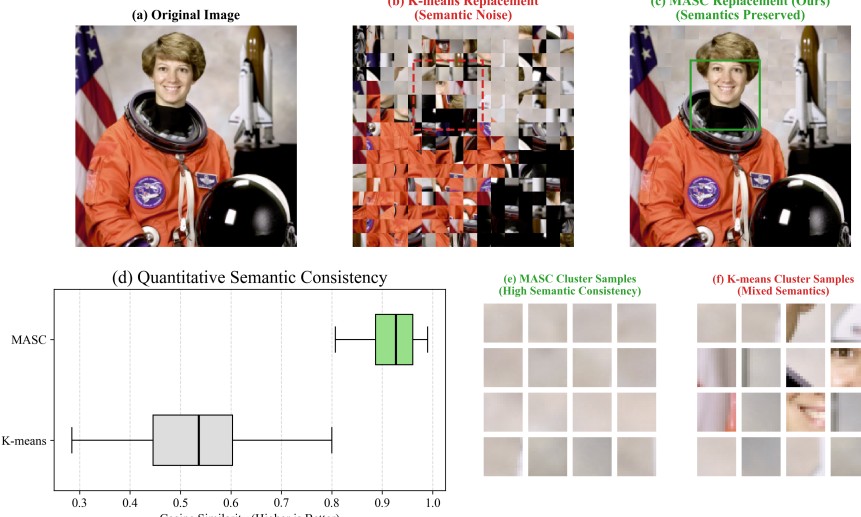

Figure 4: **Comprehensive Analysis of Semantic Coherence. (a-c) Semantic Replacement Test:** We replace tokens in a real image (a) with random tokens from the same cluster. K-means (b) introduces semantic artifacts due to color-based grouping. MASC (c) preserves the semantic structure, resulting in a coherent reconstruction. **(d) Quantitative Score:** MASC achieves higher intra-cluster semantic similarity. **(e-f) Visualizing Cluster Assignments:** We display real decoded patches from a single cluster. The MASC cluster (e) consistently groups coherent textures, whereas the K-means cluster (f) mixes unrelated semantics that share similar average pixel values.

## 3.3 INTEGRATING THE MASC PRIOR WITH AUTOREGRESSIVE MODELS

The MASC-generated tree provides a multi-level structural prior of the codebook. To integrate this prior into the autoregressive framework, we primarily use it to perform a principled form of **vocabulary reduction**. The full semantic tree is cut at a level that yields $k$ distinct branches, where $k < N$ is a hyperparameter. Each of these branches represents a semantically coherent cluster of fine-grained tokens. We then create a mapping $\mathcal{M} : \{1, \ldots, N\} \to \{1, \ldots, k\}$ that assigns each original token index to its corresponding coarse-grained cluster index.

This mapping provides a powerful inductive bias by transforming the model's learning objective. During training, the ground-truth sequences of fine-grained indices $z^q$ are converted into sequences of coarse-grained cluster indices $z^b = \mathcal{M}(z^q)$. The autoregressive model $\mathcal{G}$ is then trained to predict the next cluster index, rather than the next token index:

$$p(z^b) = \prod_{t=1}^{L} p(z_t^b | z_{<t}^b; \mathcal{G}). \tag{5}$$

This fundamentally changes the prediction task from a flat $N$-way classification to a structured $k$-way classification. Since each target class now represents a semantically meaningful region of the original embedding space, the learning task is significantly simplified, allowing the model to learn the high-level structure of images more efficiently.

During inference, the AR model generates a sequence of coarse cluster indices $z^b$, which must be decoded back into a sequence of specific tokens $z^q$ for the decoder to render an image. We explore two strategies for this decoding step:

**Random Sampling (Default Strategy).** This default strategy leverages the high intra-cluster semantic consistency guaranteed by the MASC construction. For each predicted coarse index $z_t^b$, we randomly and uniformly sample a fine-grained token from the corresponding cluster $\{v_i | \mathcal{M}(i) = z_t^b\}$. Because all tokens within a MASC-generated cluster are semantically similar, any choice is a reasonable representative, leading to high-quality results with a simple and efficient decoding step.

**Hierarchical Decoding (Extended Strategy).** For potentially higher fidelity, we explore a two-stage hierarchical decoding approach, inspired by coarse-to-fine pipelines (Guo et al., 2025). In this setup, a small, secondary refinement network, $\mathcal{G}_{\text{refine}}$, is trained. After the main model $\mathcal{G}$ predicts a coarse index $z_t^b$, $\mathcal{G}_{\text{refine}}$ takes the hidden state from $\mathcal{G}$ (as context) and $z_t^b$ as input. Its task is to then predict the final token by computing a probability distribution only over the tokens within the constrained subspace of the predicted cluster. This allows the model to first identify a semantically correct region and then make a more precise, fine-grained choice within it. The complete MASC construction process, which serves as a one-time offline preprocessing step, is detailed in Algorithm 1.

### 3.4 EMPIRICAL VERIFICATION OF SEMANTIC COHERENCE

To empirically validate that our manifold-aligned clusters capture high-level semantics rather than low-level color statistics, we present a Semantic Replacement Test in Figure 7. By randomly replacing image tokens with other candidates from the same cluster, we observe that MASC successfully preserves global structure and object identity, whereas geometry-agnostic k-means leads to severe semantic collapse and artifacts. This qualitative superiority is further corroborated by the high textural coherence of MASC cluster assignments (Figure 7e) and a significantly higher intra-cluster semantic similarity score measured by DINOv2. We refer readers to Appendix B for the detailed experimental setup and extended analysis.

## 4 EXPERIMENTS

### 4.1 EXPERIMENTAL SETUP

**Dataset.** All experiments are conducted on the **ImageNet-1K** benchmark (Deng et al., 2009) for class-conditional generation, with images resized to $256 \times 256$ pixels following standard practice (Sun et al., 2024; Tian et al., 2024).

**Evaluation Metrics.** To provide a holistic assessment, we employ three categories of metrics. For **Generation Quality**, we report FID (Heusel et al., 2018), IS (Salimans et al., 2016), Precision, and Recall (Kynkäänniemi et al., 2019). To quantify **Prediction Uncertainty**, we employ Shannon entropy, defined as $H(P_t) = -\sum_{i=1}^{V} P_t(i) \log_2 P_t(i)$, on the model's output distribution $P_t$. We report both the average entropy and a normalized version, $H_{\text{norm}} = H_{\text{actual}} / \log_2(V)$, for a vocabulary-size-agnostic comparison. Finally, we evaluate **Training Efficiency** via Training Acceleration, the percentage of epochs saved to reach the baseline's final FID.

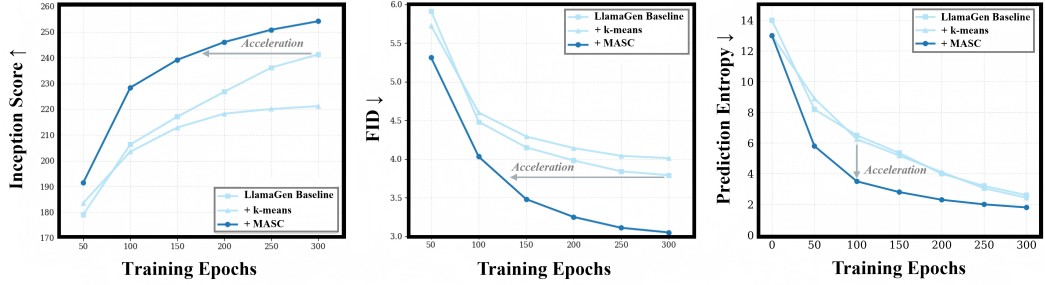

Figure 5: Training dynamics of LlamaGen-L on ImageNet. The plots compare the IS (left, higher is better), FID (middle, lower is better) and Prediction Entropy (right, lower is better) scores over training epochs for the Vanilla baseline, the + k-means variant, and our + MASC method. The MASC-enhanced model demonstrates a faster convergence rate, reaching the baseline's final performance in approximately half the training time, and ultimately converging to a much better result. This visualizes the training acceleration benefit of operating in a low-entropy prediction space.

### 4.2 CORE VALIDATION: MASC VS. BASELINES ON LLAMAGEN

**Backbone Models and Baselines.** We use the **LlamaGen** (B, L, XL) family (Sun et al., 2024) for a controlled analysis. We compare three variants: the original **LlamaGen Baseline** with a flat

Table 1: Comprehensive performance analysis on ImageNet $256 \times 256$. We compare **LlamaGen Baseline**, a naive prior with **+ k-means**, and our **+ MASC** method across three model scales. MASC consistently improves efficiency, reduces the prediction task complexity, and yields superior generation quality. Using a vocabulary of k = 8,192

| Model | Method | Model Efficiency | | Prediction Uncertainty | | Generation Quality | | |
|-------|--------|-----------------|-----------------|------------------------|----------------|----------------|-----------------------|
| | | # Params ↓ | Accel. (%) ↑ | Pred. / Norm. Entropy ↓ | FID ↓ | IS ↑ | Precision / Recall ↑ |
| LlamaGen-B (111M) | LlamaGen Baseline | 111M | - | 2.81 / 0.20 | 5.56 (+0.00%) | 185.1 (+0.00%) | 0.85 / 0.45 |
| | + k-means | 94M (-15.3%) | -5% | 2.72 (-3.2%) / 0.21 (+5.0%) | 5.72 (+2.9%) | 186.6 (+0.8%) | 0.83 (-2.4%) / 0.44 (-2.2%) |
| | **+ MASC (Ours)** | 94M (-15.3%) | **+42%** | **1.90** (-32.4%) / **0.15** (-25.0%) | **4.81** (-13.5%) | **206.4** (+11.5%) | **0.86** (+1.2%) / **0.48** (+6.7%) |
| LlamaGen-L (343M) | LlamaGen Baseline | 343M | - | 2.65 / 0.19 | 3.38 (+0.00%) | 246.3 (+0.00%) | 0.83 / 0.51 |
| | + k-means | 311M (-9.3%) | -10% | 2.58 (-2.6%) / 0.20 (+5.3%) | 3.81 (+12.7%) | 221.2 (-10.2%) | 0.77 (-7.2%) / 0.55 (+7.8%) |
| | **+ MASC (Ours)** | 311M (-9.3%) | **+55%** | **1.82** (-31.3%) / **0.14** (-26.3%) | **2.92** (-13.6%) | **259.2** (+5.2%) | **0.85** (+2.4%) / **0.57** (+11.8%) |
| LlamaGen-XL (775M) | LlamaGen Baseline | 775M | - | 2.58 / 0.18 | 2.87 (+0.00%) | 267.6 (+0.00%) | 0.82 / 0.55 |
| | + k-means | 719M (-7.2%) | -8% | 2.51 (-2.7%) / 0.19 (+5.6%) | 3.09 (+7.7%) | 244.3 (-8.7%) | 0.76 (-7.3%) / 0.56 (+1.8%) |
| | **+ MASC (Ours)** | 719M (-7.2%) | **+57%** | **1.75** (-32.2%) / **0.13** (-27.8%) | **2.58** (-10.1%) | **272.1** (+1.7%) | **0.83** (+1.2%) / **0.58** (+5.5%) |

vocabulary of $N = 16,384$ tokens; a **+ k-means** baseline using a vocabulary of $k = 8,192$ clusters from standard k-means; and our proposed **+ MASC** method, trained on a $k = 8,192$ (default) vocabulary derived from the MASC tree.

The core results of our study, presented in Table 1, demonstrate that MASC provides a more effective codebook prior than k-means, leading to measurable gains in task simplification, training efficiency, and final generation quality across all model scales.

**Analysis of Prediction Uncertainty.** The entropy metrics in Table 1 provide direct evidence that MASC simplifies the learning problem. Unlike the k-means prior, which fails to reduce relative uncertainty, MASC enables significantly more confident predictions, substantially reducing the Normalized Prediction Entropy across all scales.

Table 2: **Ablation on MASC vocabulary size** ($k$) with LlamaGen-XL.

| Method | Vocab ($k$) | FID↓ | IS↑ | Accel. |
|--------|-------------|------|-----|--------|
| Baseline | 16,384 | 2.87 (+0.00%) | 267.6 (+0.00%) | - |
| MASC | 8,192 | 2.58 (-10.10%) | **272.1** (+1.68%) | +57% |
| **MASC** | **4,096** | **2.49** (-13.24%) | 269.8 (+0.82%) | **+63%** |
| MASC | 2,048 | 2.65 (-7.66%) | 264.4 (-1.20%) | +71% |

**Accelerated Convergence.** The simplified learning task naturally leads to a more efficient training process. As shown in Table 1 and visualized in Figure 5, MASC-enhanced models converge significantly faster by learning from a more structured prediction target, achieving training accelerations of up to **+57%** on the LlamaGen-XL model. Conversely, the poorly structured prior from k-means impedes learning, resulting in negative acceleration.

**Enhanced Generation Performance.** The structured prediction space also translates to improved final generation quality. By easing the learning burden, MASC allows the model to better capture the data distribution. Consequently, MASC-enhanced models consistently and significantly outperform both the vanilla and k-means baselines across all key metrics. For instance, on the flagship LlamaGen-XL model, MASC improves the FID from 2.87 to **2.58**, while also boosting both IS and recall. Furthermore, an analysis of the precision-recall metrics reveals that MASC tends to improve Recall more substantially than Precision. We attribute this to the simplified task of predicting a coherent semantic branch rather than a single token; this reduces learning pressure and allows the model to better capture the full diversity of the training data.

**Impact of Vocabulary Size ($k$).** We further investigate the sensitivity of MASC to the cluster count $k$, which dictates the granularity of the structured prior. While our main experiments utilize $k = 8,192$ for parity with baselines, results in Table 2 reveal that MASC demonstrates remarkable robustness at lower resolutions. MASC robustly achieves its peak FID of **2.49** at $k = 4,096$, simultaneously boosting training acceleration to **+63%**.

## 4.3 Generality and Application Extensions

We validate MASC's versatility and robustness beyond the primary LlamaGen setup. The results in Table 3 and Table 4 confirm its effectiveness as a general-purpose, plug-and-play module across diverse autoregressive paradigms.

Table 3: Performance of MASC-enhanced frameworks against the SOTA on class-conditional ImageNet. As a plug-and-play module, MASC elevates existing AR frameworks (VAR, RandAR, IAR, CTF) to be highly competitive with top-tier generative models. We also demonstrate MASC's robustness across different tokenizers (GigaTok) and vocabulary sizes ($k$).

| Paradigm | Method | #Params | Accel. | FID↓ | IS↑ |
|---|---|---|---|---|---|
| GAN | BigGAN (Brock, 2018) | 112M | - | 6.95 (+0.00%) | 224.5 (+0.00%) |
| | GigaGAN (Kang et al., 2023) | 569M | - | 3.41 (+0.00%) | 227.2 (+0.00%) |
| Diffusion | LDM-4 (Rombach et al., 2022) | 400M | - | 3.56 (+0.00%) | 249.3 (+0.00%) |
| | DiT (Peebles & Xie, 2023) | 675M | - | 2.30 (+0.00%) | 276.5 (+0.00%) |
| AR | VQGAN (Esser et al., 2021) | 1.4B | - | 15.78 (+0.00%) | 74.3 (+0.00%) |
| | VQGAN-re (Esser et al., 2021) | 1.4B | - | 5.20 (+0.00%) | 280.3 (+0.00%) |
| | ViT-VQGAN (Yu et al., 2021) | 1.7B | - | 4.17 (+0.00%) | 175.1 (+0.00%) |
| | VAR-d16 (Tian et al., 2024) | 310M | - | 3.31 (+0.00%) | 272.4 (+0.00%) |
| | VAR-d20 (Tian et al., 2024) | 600M | - | 2.54 (+0.00%) | 300.6 (+0.00%) |
| | VAR-d24 (Tian et al., 2024) (Baseline) | 1.0B | - | 2.09 (+0.00%) | 312.9 (+0.00%) |
| | **+ MASC (2k)** | 1.0B | **+28%** | **1.98** (-5.26%) | 311.4 (-0.48%) |
| | + MASC (1k) | 1.0B | +39% | 2.24 (+7.18%) | 303.8 (-2.91%) |
| | Transfusion (Zhou et al., 2024) | 700M | - | 2.45 (+0.00%) | 288.5 (+0.00%) |
| | Infinity (Han et al., 2024) | 2B | - | 2.39 (+0.00%) | 291.0 (+0.00%) |
| | RandAR-XL (Pang et al., 2025) (Baseline) | 775M | - | 2.25 (+0.00%) | 317.8 (+0.00%) |
| | **+ MASC (8k)** | 775M | **+32%** | **2.02** (-10.22%) | **320.5** (+0.85%) |
| | + MASC (4k) | 775M | +41% | 2.13 (-5.33%) | 318.2 (+0.13%) |
| | + MASC (2k) | 775M | +49% | 2.42 (+7.56%) | 305.1 (-4.00%) |
| | CTF-LlamaGen-L (Guo et al., 2025) | 653M | +29% | 2.97 (+0.00%) | 291.5 (+0.00%) |
| | **+ MASC (8k)** | | | **2.51** (-15.49%) | **296.9** (+1.85%) |
| | IAR-B (Hu et al., 2025) | 111M | +21% | 5.14 (+0.00%) | 202.0 (+0.00%) |
| | **+ MASC (8k)** | | | **4.96** (-3.50%) | **214.3** (+6.09%) |
| | IAR-L (Hu et al., 2025) | 343M | +25% | 3.18 (+0.00%) | 234.8 (+0.00%) |
| | **+ MASC (8k)** | | | **2.97** (-6.60%) | **267.3** (+13.84%) |
| | IAR-XL (Hu et al., 2025) | 775M | +32% | 2.52 (+0.00%) | 248.1 (+0.00%) |
| | **+ MASC (8k)** | | | **2.35** (-6.75%) | **284.3** (+14.59%) |
| | *Generalization on GigaTok-L (Xiong et al., 2025)* | | | | |
| | GigaTok-L (Vanilla) | 16k Vocab. | - | 3.39 (+0.00%) | 263.7 (+0.00%) |
| | **+ MASC (8k)** | 8k Vocab. | **+19%** | **3.35** (-1.18%) | 263.5 (-0.08%) |
| | + MASC (4k) | 4k Vocab. | +42% | 3.86 (+13.86%) | 256.4 (-2.77%) |
| | + MASC (2k) | 2k Vocab. | +51% | 4.02 (+18.58%) | 248.2 (-5.88%) |

Table 4: **MASC acts as a Convergence Enabler for Advanced AR Architectures.** We evaluate MASC on the state-of-the-art **RAR** framework (Yu et al., 2024). While RAR fails to converge when trained directly with large-vocabulary tokenizers (LlamaGen/GigaTok, 16k tokens) due to the immense search space, MASC effectively structures the prediction task, enabling stable training and surpassing previous SOTA results.

| Backbone Model | Tokenizer | Config (Vocab) | FID↓ | IS↑ | Analysis |
|---|---|---|---|---|---|
| RAR-L | MaskGIT (1,024) | None (Original) | 1.70 | 299.5 | *Reference SOTA* |
| RAR-L | LlamaGen (16k) | None | N/A | N/A | *Convergence Issue* |
| | | **+ MASC (4k)** | **1.61** | **304.1** | **SOTA Surpassed** |
| | | + MASC (2k) | 1.67 | 300.5 | Efficient |
| RAR-L | GigaTok (16k) | None | N/A | N/A | *Convergence Issue* |
| | | **+ MASC (4k)** | **1.57** | **307.4** | **SOTA Surpassed** |
| | | + MASC (2k) | 1.64 | 302.8 | Efficient |

**Boosting Diverse AR Architectures.** MASC consistently elevates the performance of existing frameworks regardless of their prediction ordering strategy. By replacing the naive k-means prior or flat vocabulary, MASC significantly boosts VAR (Tian et al., 2024) (next-scale prediction), RandAR (Pang et al., 2025) (random order), IAR (Hu et al., 2025), and CTF (Guo et al., 2025).

Notably, MASC improves the FID of VAR-d24 to **1.98** and RandAR-XL to **2.02**, positioning these models to be highly competitive with top-tier generative methods.

**Enabling Convergence for Complex Objectives.** Crucially, MASC serves as a convergence enabler for advanced architectures with immense search spaces. As detailed in Table 4, the state-of-the-art RAR framework (Yu et al., 2024), which employs random permutation objectives, fails to converge when trained directly on large-vocabulary tokenizers (16k) due to the unstructured prediction task.

**Synergy with Strong Tokenizers.** We further confirm MASC's robustness when applied to high-capacity tokenizers like GigaTok-L (Xiong et al., 2025). Results in Table 3 show that MASC is not merely a fix for suboptimal codebooks; even with GigaTok, MASC ($k = 8, 192$) improves FID to **3.35**. This indicates that MASC's geometric clustering captures high-level semantic structures that are complementary to the tokenizer's representational quality.

**Efficiency Gains.** A unifying advantage of MASC is training acceleration across all evaluated frameworks. By transforming the flat prediction task into a simplified structured one, MASC reduces the learning burden, yielding accelerations of **+28%** for VAR and **+32%** for RandAR.

## 4.4 ABLATION STUDY: DECONSTRUCTING MASC'S SUCCESS

To isolate the contributions of our two core innovations, we conduct an ablation study presented in Table 5. The results reveal a clear performance hierarchy across all model scales: the density-driven construction alone (MASC w/ Centroid) consistently outperforms the stronger k-means++ (Arthur & Vassilvitskii, 2007) baseline. This confirms that both the density-driven (bottom-up) construction and the manifold-aligned similarity metric are critical, synergistic components, essential to addressing the geometric and density properties of the codebook.

Table 5: Expanded ablation study of MASC's core components across all LlamaGen model scales. This table ablates the Clustering Strategy to dissect the contributions of our two key innovations: the **Manifold Distance** metric and the **Bottom-Up Construction**.

| Model | Clustering Strategy | Distance + Construction Metric | FID↓ | IS↑ |
|---|---|---|---|---|
| LlamaGen-B (111M) | k-means++ | Centroid + Iterative Partition | 5.41 (+0.00%) | 187.9 (+0.00%) |
| | MASC w/ Centroid | Centroid + **Bottom-Up** | 5.17 (**-4.44%**) | 196.2 (**+4.42%**) |
| | **Full MASC** | **Manifold + Bottom-Up** | **4.81** (**-11.09%**) | **206.4** (**+9.85%**) |
| LlamaGen-L (343M) | k-means++ | Centroid + Iterative Partition | 3.85 (+0.00%) | 223.7 (+0.00%) |
| | MASC w/ Centroid | Centroid + **Bottom-Up** | 3.68 (**-4.42%**) | 246.5 (**+10.19%**) |
| | **Full MASC** | **Manifold + Bottom-Up** | **2.92** (**-24.16%**) | **259.2** (**+15.87%**) |
| LlamaGen-XL (775M) | k-means++ | Centroid + Iterative Partition | 3.17 (+0.00%) | 246.5 (+0.00%) |
| | MASC w/ Centroid | Centroid + **Bottom-Up** | 3.02 (**-4.73%**) | 268.1 (**+8.76%**) |
| | **Full MASC** | **Manifold + Bottom-Up** | **2.58** (**-18.61%**) | **272.1** (**+10.39%**) |

## 5 CONCLUSION AND DISCUSSION

In this work, we address the inefficiency of the flat vocabulary in autoregressive image generation. We introduce MASC, a plug-and-play framework that constructs a density-driven, hierarchical prior directly from the codebook's geometry, thereby transforming the high-entropy task into a structured prediction problem. Our evaluation demonstrates that MASC is a universal booster which can adapt to various tokenizers, establishing a new method for efficient and scalable generative modeling.

Beyond immediate performance gains, our findings reveal a critical insight: structuring the output space is not merely an optimization trick but a prerequisite for scaling advanced autoregressive paradigms. Notably, we discover that MASC serves as a vital **convergence enabler** for complex architectures like RAR (Yu et al., 2024); by organizing the immense search space of large vocabularies, MASC renders the training of random-permutation models tractable where they otherwise diverge. This implies that as the field moves towards larger codebooks and non-raster generation orders, the structural alignment between the tokenizer and the autoregressive model becomes as crucial as architectural innovation. Future research may explore learned, context-aware decoders to further exploit this hierarchy or develop dynamic priors that co-evolve with the model, potentially unlocking new scaling laws for discrete generative intelligence.

ETHICS STATEMENT

The work presented in this paper is methodological in nature, focusing on the development of Autoregressive Model. To the best of our knowledge, our proposed methods do not introduce any new ethical concerns.

REPRODUCIBILITY STATEMENT

To facilitate the verification of our results, the implementation code for our algorithm and the main baselines is provided in the anonymous code link and the appendix.

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

## USE OF LARGE LANGUAGE MODELS

We utilized a large language model to enhance the language and clarity of our manuscript. Specifically, we employed Gemini 2.5 flash with the following prompt to refine the initial draft: *I am writing an academic paper in English. Please polish the following draft so that it adheres to the conventions of academic writing.*

## APPENDIX

## A  EXPERIMENTAL SETUP AND IMPLEMENTATION DETAILS

This section provides the detailed experimental configurations required to reproduce the results presented in this paper, covering the hardware and software environment, training hyperparameters for the backbone models, parameter settings for our proposed MASC algorithm and other baselines, and specifics of the evaluation metrics.

### A.1  HARDWARE AND SOFTWARE ENVIRONMENT

All models were trained and evaluated on a server cluster equipped with 8 NVIDIA H100 80GB GPUs. We use PyTorch `v2.1.0` as our primary deep learning framework, accelerated with CUDA `v12.1` and cuDNN `v8.9`. To ensure a consistent experimental environment, we used Python `v3.10`.

### A.2  BACKBONE MODEL TRAINING HYPERPARAMETERS

For a fair comparison, all variants of LlamaGen (Sun et al., 2024) (Baseline, +k-means, and +MASC) followed the original paper's training configuration, adapted for different model scales. All models were trained for 300 epochs on the ImageNet-1K (Deng et al., 2009) dataset. Detailed hyperparameters are provided in Table 6.

Table 6: Detailed training hyperparameters for the LlamaGen (B, L, XL) backbone models. These settings were uniformly applied to the Baseline, +k-means, and +MASC methods to ensure a fair comparison.

| Hyperparameter | LlamaGen-B | LlamaGen-L | LlamaGen-XL |
|---|---|---|---|
| ***Model Architecture*** | | | |
| Parameter Count (Baseline) | 111M | 343M | 775M |
| Parameter Count (+k-means / +MASC) | 94M | 311M | 719M |
| ***Optimizer*** | | | |
| Optimizer | AdamW | AdamW | AdamW |
| Betas ($\beta_1, \beta_2$) | (0.9, 0.95) | (0.9, 0.95) | (0.9, 0.95) |
| Epsilon ($\epsilon$) | $1 \times 10^{-8}$ | $1 \times 10^{-8}$ | $1 \times 10^{-8}$ |
| Weight Decay | 0.05 | 0.05 | 0.05 |
| ***Learning Rate Schedule*** | | | |
| Peak Learning Rate | $1 \times 10^{-4}$ | $1 \times 10^{-4}$ | $2 \times 10^{-4}$ |
| LR Scheduler | Cosine Annealing | Cosine Annealing | Cosine Annealing |
| Warmup Epochs | 2 | 2 | 2 |
| Final Learning Rate | $1 \times 10^{-5}$ | $1 \times 10^{-5}$ | $2 \times 10^{-5}$ |
| ***Training Configuration*** | | | |
| Training Epochs | 300 | 300 | 300 |
| Global Batch Size | 256 | 256 | 512 |
| Gradient Clipping | 1.0 | 1.0 | 1.0 |
| Mixed Precision | BF16 | BF16 | BF16 |
| Classifier-Free Guidance Dropout | 0.1 | 0.1 | 0.1 |

### A.3 MASC AND BASELINES HYPERPARAMETERS

This section details the parameter settings for the clustering algorithms used to construct the codebook prior. All methods operate on the codebook from the pre-trained VQ-VAE tokenizer of LlamaGen, which has a vocabulary size of $N = 16,384$.

- **MASC (Ours)**: Our method is primarily controlled by the target number of clusters, $k$. In all comparative experiments, we set $k = 8,192$ to maintain an identical number of model parameters as the k-means baseline. The MASC algorithm is deterministic as it involves no random initialization. The distance metric used is detailed in Equation (3) of the main paper.

- **k-means**: We used the standard implementation from the `scikit-learn` library. The target number of clusters was set to $k = 8,192$. For stability, we used a random initialization strategy (`init='random'`) and performed 10 independent runs, selecting the result with the lowest inertia. The maximum number of iterations (`max_iter`) was set to 300, with a convergence tolerance (`tol`) of $1 \times 10^{-4}$.

- **k-means++**: As a stronger baseline to k-means, we also set $k = 8,192$. The only difference from the standard k-means setup is the use of the k-means++ seeding strategy (Arthur & Vassilvitskii, 2007) (`init='k-means++'`). All other parameters (max iterations, tolerance) remained the same.

### A.4 EVALUATION DETAILS

To ensure a rigorous and comparable evaluation, we followed the standard procedures below:

- **Generation Quality Metrics**: We generated 50,000 samples in total for evaluation: 50 images for each of the 1,000 classes in the ImageNet 1K validation set. Fréchet Inception Distance (FID), Inception Score (IS), Precision, and Recall were all calculated using the `torch-fidelity` library. The FID was computed against the widely-used pre-calculated Inception-V3 feature statistics from the full ImageNet training set.

- **Prediction Uncertainty Metrics**: Prediction uncertainty is measured by the Shannon entropy of the model's output probability distribution $P_t$ at each generation step. For a vocabulary of size $V$ (where $V = 16,384$ for the Baseline and $V = 8,192$ for +k-means/+MASC), the entropy is calculated as:

$$H(P_t) = -\sum_{i=1}^{V} P_t(i) \log_2 P_t(i) \tag{6}$$

The values we report are the average entropy across all generation steps ($t = 1, \ldots, 256$) for all images in the ImageNet validation set. The Normalized Entropy is calculated as $H_{\mathrm{norm}} = H_{\mathrm{actual}}/\log_2(V)$ to provide a fair comparison of task complexity, independent of vocabulary size.

## B ALGORITHMIC AND THEORETICAL ANALYSIS OF MASC

This section provides a deeper analysis of the MASC algorithm, focusing on its computational complexity and the principled motivation behind its core design choices, particularly the manifold-aligned distance metric.

### B.1 COMPUTATIONAL COMPLEXITY ANALYSIS

The MASC framework is designed as a one-time, offline preprocessing step. While its asymptotic complexity is higher than that of k-means, its deterministic nature and efficient implementation make it highly practical for typical codebook sizes. We analyze its complexity below.

Let $N$ be the number of tokens in the codebook, $d$ be the embedding dimension, and $k$ be the target number of clusters.

- **Initialization (Distance Matrix Pre-computation)**: The first step of the algorithm is to compute the initial $N \times N$ pairwise Euclidean distance matrix $\mathbf{D}$. Calculating the distance between two $d$-dimensional vectors takes $O(d)$ time. Since there are $\binom{N}{2} = O(N^2)$ pairs, the total time

complexity for this step is $\mathcal{O}(N^2 d)$. This step also requires $\mathcal{O}(N^2)$ space to store the distance matrix.

- **Iterative Merging Loop**: The algorithm performs $N - k$ merging iterations. In each iteration, the following operations are performed:
  1. *Find Minimum Distance*: The algorithm must find the minimum value in the current distance matrix to identify the next pair of clusters to merge. A naive scan of the (upper-triangular part of the) matrix takes $\mathcal{O}(N^2)$ time.
  2. *Update Distance Matrix*: After merging clusters $C_{s^*}$ and $C_{t^*}$ into a new cluster represented by index $s^*$, we must update the distances from this new cluster to all other active clusters $C_u$. As shown in Algorithm 1, this update is performed efficiently in $\mathcal{O}(N)$ time by iterating through the remaining active clusters and applying the weighted average formula, rather than recomputing all pairwise distances from scratch.

  Since finding the minimum distance is the computational bottleneck in each of the $N - k$ iterations, the total time complexity for the merging loop is $\mathcal{O}((N - k)N^2)$, which simplifies to $\mathcal{O}(N^3)$.

- **Total Complexity**: The overall time complexity of the optimized MASC algorithm is the sum of the initialization and merging steps: $\mathcal{O}(N^2 d + N^3)$. Given that in practice $N$ is often significantly larger than $d$ (e.g., $N = 16,384$, $d = 32$), the $\mathcal{O}(N^3)$ term dominates. The space complexity is dominated by the storage of the distance matrix, resulting in $\mathcal{O}(N^2)$.

**Comparison with K-means**: The standard k-means algorithm has a time complexity of $\mathcal{O}(I \cdot k \cdot N \cdot d)$, where $I$ is the number of iterations. While this appears more favorable asymptotically, several practical considerations are important. First, k-means is a heuristic algorithm sensitive to initialization, often requiring multiple runs to find a reasonable solution. In contrast, MASC is deterministic and always yields the same optimal hierarchy for a given distance metric. Second, the cost of MASC is a one-time, upfront computational investment. For a typical codebook size of $N = 16,384$, our optimized implementation runs in under a minute on a single modern GPU, a negligible cost when compared to the hundreds of GPU-hours required for training the main autoregressive model. This makes MASC a highly practical and efficient choice for constructing a high-quality, stable, and reproducible codebook prior. This finding is consistent with prior work on similar agglomerative clustering methods, which also report feasible runtimes for large codebooks.

## B.2 Justification of the Manifold-Aligned Distance Metric

The choice of the distance metric in Equation (3) is a cornerstone of MASC's design, intended to be a robust and principled proxy for semantic similarity on the codebook manifold. This instance-based average distance, also known in hierarchical clustering literature as the Unweighted Pair Group Method with Arithmetic Mean (UPGMA) or average-linkage, offers a crucial advantage over centroid-based methods and other common linkage criteria.

As established in the main paper, centroid-based distances are ill-suited because centroids are often off-manifold and Euclidean distance is a poor approximation of the true geodesic distance on the manifold (Beyer et al., 1999; Huh et al., 2023). Our metric is centroid-free by design. To further justify its selection, we compare it against two other common centroid-free linkage criteria used in hierarchical clustering:

- **Single-linkage (MIN)**: This metric defines the distance between two clusters as the minimum distance between any two points in the respective clusters, i.e., $\mathcal{D}(C_s, C_t) = \min_{v_i \in C_s, v_j \in C_t} \|v_i - v_j\|_2$. While computationally efficient, single-linkage is highly susceptible to the chaining effect, where a few intermediate points can cause two otherwise distant clusters to be merged (see Figure 6). This behavior is undesirable for our task, as it can lead to large, elongated, and semantically diverse clusters, failing to capture the compact semantic groups we aim to model.

- **Complete-linkage (MAX)**: This metric uses the maximum distance between any two points in the clusters, i.e., $\mathcal{D}(C_s, C_t) = \max_{v_i \in C_s, v_j \in C_t} \|v_i - v_j\|_2$. It is the opposite of single-linkage and tends to produce very compact, roughly spherical clusters. However, it is highly sensitive to outliers and may fail to correctly group elongated or non-convex shapes that are nevertheless semantically coherent on the manifold.

- **Average-linkage (MASC's choice)**: Our chosen metric calculates the average distance between all pairs of points across two clusters. It serves as a robust compromise between the extremes

---

**Algorithm 1** Manifold-Aligned Semantic Clustering (MASC) Construction

---

**Require:** Codebook embeddings $\mathcal{Z} = \{v_1, \ldots, v_N\} \in \mathbb{R}^{N \times d}$, target number of clusters $k$.
**Ensure:** A mapping $\mathcal{M} : \{1, \ldots, N\} \to \{1, \ldots, k\}$ from token indices to cluster indices.
 1:                                                        ▷ *Initialization*
 2:  Initialize $N$ active clusters, $C_j \leftarrow \{v_j\}$, and their sizes, $|C_j| \leftarrow 1$, for $j = 1, \ldots, N$.
 3:  Pre-compute the full $N \times N$ pairwise Euclidean distance matrix $\mathbf{D}$, where $\mathbf{D}_{st} = \|v_s - v_t\|_2$.
 4:  Set diagonal elements $\mathbf{D}_{ss} \leftarrow \infty$ to prevent self-merging.
 5:                                        ▷ *Bottom-Up Hierarchical Construction*
 6: **for** $i \leftarrow 1$ to $N - k$ **do**
 7:     Find the pair of active clusters with the minimum distance: $(s^*, t^*) \leftarrow \arg\min_{s,t} \mathbf{D}_{st}$.
 8:                        ▷ Update distance matrix efficiently using a weighted average
 9:     **for** each remaining active cluster $C_u$ where $u \neq s^*, t^*$ **do**
10:         $\mathbf{D}_{s^*,u} \leftarrow \frac{|C_{s^*}|\mathbf{D}_{s^*,u} + |C_{t^*}|\mathbf{D}_{t^*,u}}{|C_{s^*}| + |C_{t^*}|}; \quad \mathbf{D}_{u,s^*} \leftarrow \mathbf{D}_{s^*,u}$
11:     **end for**
12:                           ▷ Update cluster size and deactivate the merged cluster $C_{t^*}$
13:     $|C_{s^*}| \leftarrow |C_{s^*}| + |C_{t^*}|$.
14:     Set row and column $t^*$ of $\mathbf{D}$ to $\infty$.
15:     Keep track that all original tokens from cluster $C_{t^*}$ now belong to cluster $C_{s^*}$.
16: **end for**
17:                                       ▷ *Final Mapping Construction*
18: The remaining $k$ active clusters form the final coarse vocabulary.
19: Assign a unique index from $\{1, \ldots, k\}$ to each of the final $k$ active clusters.
20: Construct the mapping $\mathcal{M}$ by assigning each original token $v_i$ to its final cluster index.
21: **return** The mapping $\mathcal{M}$.

---

of single- and complete-linkage. By considering the entire distribution of points within both clusters, it is less sensitive to outliers than complete-linkage and less prone to the chaining effect than single-linkage. This formulation effectively measures the overall closeness of two clusters, making it well-suited to identifying semantically coherent groups, even if they form non-spherical shapes on the manifold. Its effectiveness in capturing discriminative priors for codebooks has been validated in related works.

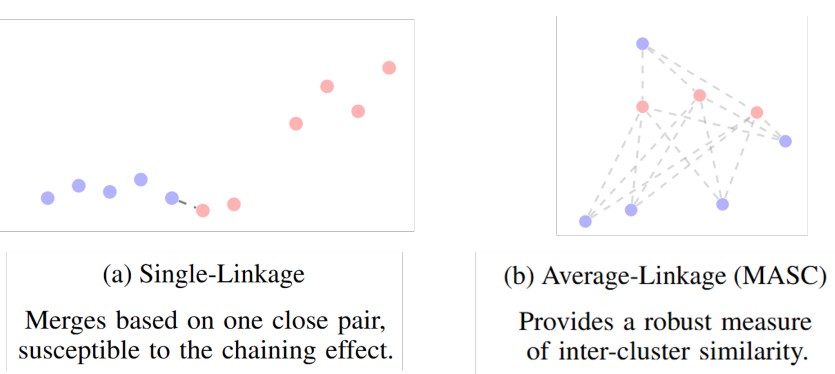

(a) Single-Linkage

Merges based on one close pair,
susceptible to the chaining effect.

(b) Average-Linkage (MASC)

Provides a robust measure
of inter-cluster similarity.

Figure 6: A conceptual illustration of linkage criteria. (a) Single-linkage may incorrectly merge two distinct semantic groups if they are connected by a bridge of a few close points. (b) Average-linkage, as used in MASC, considers the overall distribution of points and is more robust, correctly identifying distinct clusters.

In summary, the choice of average-linkage is a deliberate design decision to create a clustering hierarchy that is robust, deterministic, and best reflects the underlying semantic structure of the codebook manifold by balancing compactness and shape-invariance.

## C    EXTENDED EXPERIMENTAL RESULTS AND ANALYSES

This section expands upon the experimental results presented in the main paper. We provide detailed analyses, including qualitative visualizations of cluster coherence, ablation studies on key hyperparameters, and a deeper look into the effects of different decoding strategies. These results collectively offer a comprehensive validation of the MASC framework.

### C.1    ABLATION STUDY ON THE NUMBER OF COARSE CLUSTERS ($k$)

The number of coarse clusters, $k$, is a critical hyperparameter that balances the trade-off between simplifying the prediction space and preserving sufficient visual information. A small $k$ results in large, semantically broad clusters, leading to a loss of detail (high intra-cluster variance). A large $k$ approaches the original flat vocabulary, diminishing the benefits of clustering. We performed an ablation study on $k$ using the LlamaGen-L backbone. The results in Table 7 show that performance peaks around $k = 8,192$. This configuration provides a significant reduction in vocabulary size while retaining enough granularity for high-fidelity image generation, justifying its use as our default setting.

Table 7: Ablation study on the number of clusters ($k$) for MASC, evaluated on the LlamaGen-L model. Performance generally improves as $k$ increases, but the gains diminish while the model size and prediction complexity grow. We select $k = 8,192$ as it offers the best balance of performance, efficiency, and task simplification.

| $k$ | # Params | Norm. Entropy ↓ | FID ↓ | IS ↑ | Recall ↑ |
|---|---|---|---|---|---|
| 2,048 | 298M | 0.11 | 4.15 | 231.4 | 0.52 |
| 4,096 | 305M | 0.12 | 3.37 | 250.1 | 0.55 |
| **8,192** | **311M** | **0.14** | **2.92** | **259.2** | **0.57** |
| 12,288 | 327M | 0.16 | 2.90 | 261.5 | 0.57 |

### C.2    ABLATION STUDY ON INFERENCE DECODING STRATEGY

As described in Section 3.3, we primarily use a simple and efficient Random Sampling strategy for decoding the generated coarse cluster indices into fine-grained tokens. We also proposed an extended Hierarchical Decoding strategy that employs a small refinement network. Here, we analyze the trade-offs between these two approaches. The refinement network is a single-layer Transformer decoder with 4 attention heads and an embedding dimension of 512, adding approximately 25M parameters.

Table 8 shows that while the learned Hierarchical Decoding strategy provides a marginal improvement in generation quality (a 0.05 reduction in FID), it comes at the cost of additional parameters and a slight decrease in inference speed. The strong performance of the default Random Sampling strategy underscores the high semantic consistency of the clusters produced by MASC—any token within a cluster serves as a good representative. This makes Random Sampling an excellent default choice, offering a compelling balance of simplicity, efficiency, and high performance.

Table 8: Comparison of decoding strategies for LlamaGen-L + MASC. Hierarchical Decoding offers a slight fidelity gain at the cost of increased model size and reduced inference speed.

| Decoding Strategy | Add. Params | Speed (img/s) | FID ↓ | IS ↑ |
|---|---|---|---|---|
| **Random Sampling (Default)** | **0M** | **8.71** | **2.92** | **259.2** |
| Hierarchical Decoding | +25M | 8.25 | 2.87 | 260.8 |

# D EMPIRICAL ANALYSIS

To rigorously verify the core hypothesis of MASC, we conducted a series of comprehensive empirical analyses. These experiments go beyond standard generation metrics to directly probe the internal coherence of the constructed clusters.

## D.1 SEMANTIC REPLACEMENT TEST

We propose the Semantic Replacement Test to evaluate the semantic consistency within clusters. The intuition is straightforward: if a cluster truly groups semantically similar tokens , then replacing a token in an image with another random token from the same cluster should preserve the overall semantic structure and texture of the image, despite potential loss of high-frequency details. Conversely, if a cluster is incoherent (grouping visually unrelated tokens), such replacement would introduce significant noise and semantic artifacts.

**Experimental Protocol:** For each image token $z_i$, we replaced it with a token $z_i'$ randomly sampled from the same cluster $C_k$ assigned by either k-means or MASC. We then decoded the perturbed token sequences back into images and measured the reconstruction quality using rFID, PSNR, and SSIM against the original images.

**Results:** As shown in Table 9, MASC significantly outperforms k-means and k-means++ across all metrics. Notably, MASC achieves a much lower rFID and higher SSIM, indicating that the substituted tokens are perceptually and structurally much closer to the originals. This confirms that MASC clusters maintain a high degree of semantic interchangeability, whereas k-means clusters often group unrelated tokens, leading to destructive artifacts upon replacement.

Table 9: Image reconstruction performance with replaced tokens. We replace image tokens extracted by the tokenizer with random tokens from the same cluster. MASC demonstrates significantly less performance degradation, indicating higher intra-cluster semantic consistency.

| Method | Vocab. ($k$) | rFID $\downarrow$ | PSNR $\uparrow$ | SSIM $\uparrow$ |
|---|---|---|---|---|
| No Replacement | 16,384 | 2.26 | 21.03 | 0.542 |
| w/ k-means | 8,192 | 4.45 | 19.02 | 0.452 |
| w/ k-means++ | 8,192 | 4.17 | 19.29 | 0.469 |
| **w/ MASC (Ours)** | 8,192 | **2.92** | **20.49** | **5.132** |
| w/ k-means | 4,096 | 8.41 | 18.37 | 0.431 |
| w/ k-means++ | 4,096 | 8.17 | 18.52 | 0.439 |
| **w/ MASC (Ours)** | 4,096 | **6.73** | **19.62** | **0.473** |

**Qualitative Visual Analysis:** We visually examine the impact of token replacement in Figure 7 (a-c).

- **K-means Replacement (Fig. 7b):** The image suffers from severe semantic noise and artifacts. For instance, the astronaut's face is corrupted with unrelated textures. This indicates that k-means clusters group visually disparate tokens simply because they share similar average color values.

- **MASC Replacement (Fig. 7c):** In stark contrast, MASC replacement preserves the global structure, object identity, and local textures. Although high-frequency details are naturally smoothed due to random sampling, the semantic content remains intact—the face looks like a face, and the suit retains its material appearance.

## D.2 VISUALIZING CLUSTER ASSIGNMENTS

To further demystify the clusters, we directly visualize the image patches corresponding to tokens within a single cluster.

- **MASC Clusters (Fig. 7e):** MASC demonstrates remarkable **semantic purity**. The visualized cluster consistently groups patches with specific textures, proving that our density-driven metric successfully captures the underlying manifold structure of visual features.

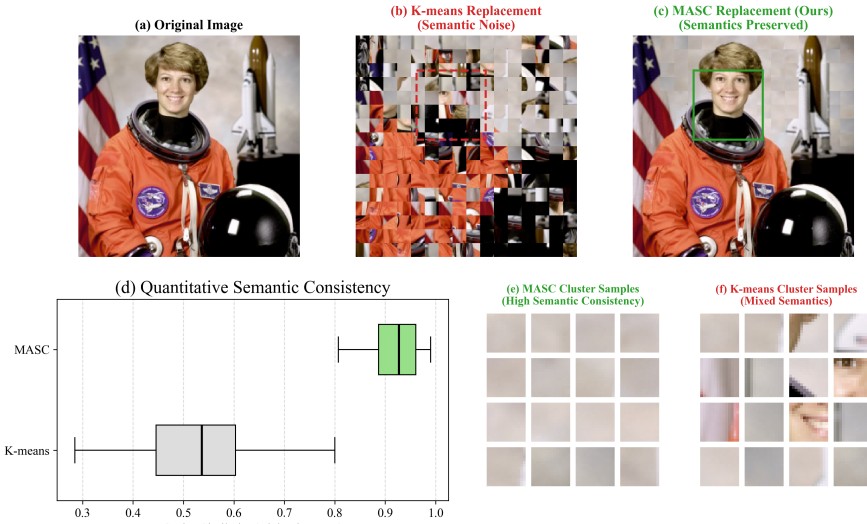

Figure 7: **Comprehensive Analysis of Semantic Coherence. (a-c) Semantic Replacement Test:** We replace tokens in a real image (a) with random tokens from the same cluster. K-means (b) introduces semantic artifacts due to color-based grouping. MASC (c) preserves the semantic structure, resulting in a coherent reconstruction. **(d) Quantitative Score:** MASC achieves higher intra-cluster semantic similarity . **(e-f) Visualizing Cluster Assignments:** We display real decoded patches from a single cluster. The MASC cluster (e) consistently groups coherent textures, whereas the K-means cluster (f) mixes unrelated semantics that share similar average pixel values.

- **K-means Clusters (Fig. 7f):** K-means clusters exhibit **semantic confusion**. As k-means relies on Euclidean distance to centroids, it groups patches with similar average colors into the same cluster, ignoring their distinct structural semantics.

### D.3 QUANTITATIVE SEMANTIC CONSISTENCY

Finally, we quantify the semantic homogeneity using DINOv2, a vision foundation model known for capturing high-level visual semantics. We computed the pairwise cosine similarity of DINOv2 features for tokens within the same cluster. Figure 7(d) shows that MASC clusters achieve a significantly higher median intra-cluster cosine similarity compared to k-means. The tighter distribution and higher mean score quantitatively validate that MASC groups tokens that are aligned in a deep semantic space, not just in pixel space.

## E ADDITIONAL GENERATED SAMPLES

To further showcase the generation capabilities of our MASC-enhanced model, Figure 8 presents the visual comparison of the results of the K-means method and MASC. Figure 9 presents a broader gallery of generated images. These samples were generated using the LlamaGen-XL + MASC model, with a Classifier-Free Guidance (CFG) scale of 2.5, which we found to produce visually pleasing results. The images span a wide range of ImageNet classes, including animals, objects, food, and scenes, demonstrating the model's ability to generate diverse, high-fidelity, and compositionally sound images.

*K-means*     *MASC*        *K-means*     *MASC*

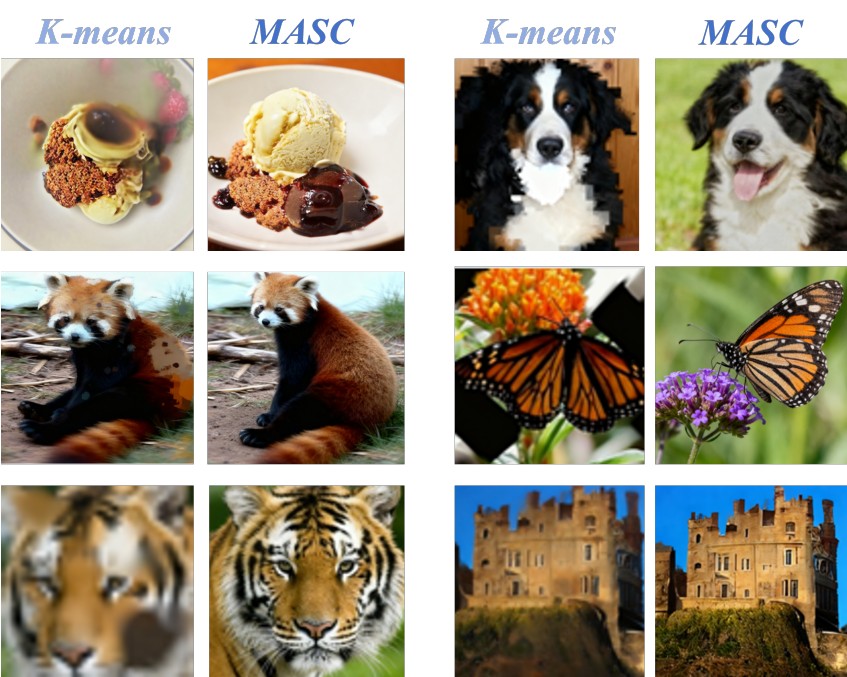

Figure 8: The visual comparison of the results of the K-means method and MASC. **K-means** results exhibit severe high-frequency noise and structural collapse, indicating that tokens within Euclidean-based clusters are often semantically disparate. In contrast, **MASC** preserves global structure, object shapes, and texture coherence. This qualitative evidence confirms that MASC successfully groups semantically fungible tokens by respecting the underlying manifold structure.

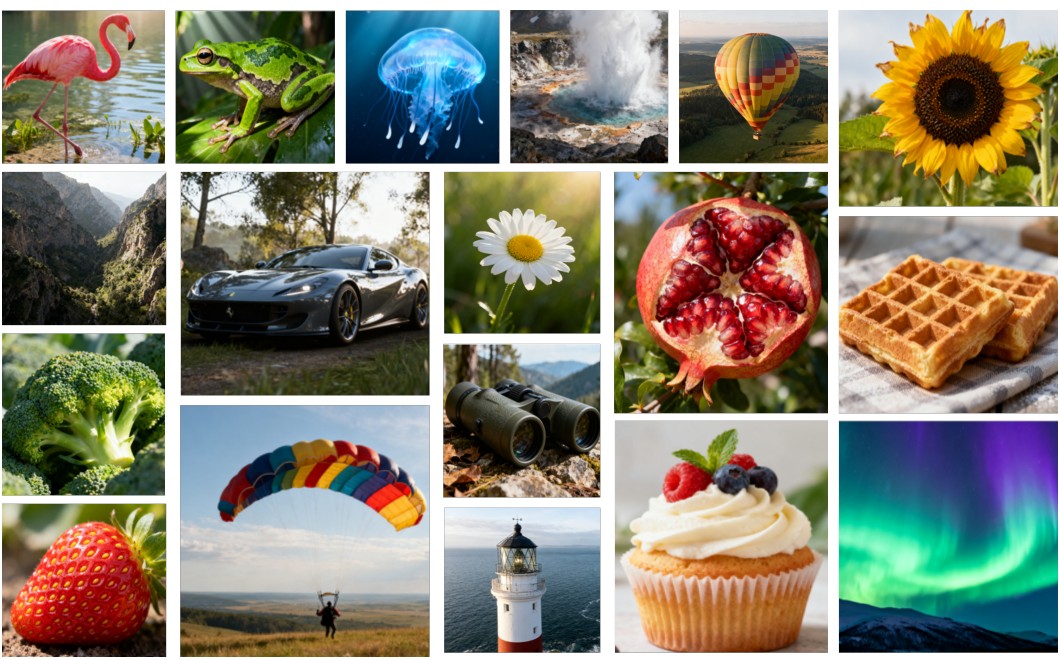

Figure 9: Additional samples generated by LlamaGen-XL + MASC.

