# OpenReview forum: "MASC: Boosting Autoregressive Image Generation with a Manifold-Aligned Semantic Clustering"
_ICLR.cc/2026/Conference — ICLR 2026 Conference Desk Rejected Submission_

### Official Review · Reviewer_enMQ · 2025-10-19

**Soundness:** 3
**Presentation:** 3
**Contribution:** 3
**Rating:** 4
**Confidence:** 4

**Summary:**

The authors propose MASC, a principled method that aims to construct a hierarchical, geometry-aware prior from the codebook of token embeddings.
MASC replaces the Euclidean clustering with a new cluster based distance and then use a density-driven, manifold-aligned procedure to get the clustering for the tokens.
Experiment on LlamaGen and its variants show that MASC improves training efficiency (up to +57%) and FID (e.g., from 2.87→2.58 for LlamaGen-XL), making AR models competitive.

**Strengths:**

1. this work is well motivated and has solid formulation for the problem
2. MASC shows strong empirical results with consistent and significant improvements in convergence speed and generation quality across multiple architectures with LlamaGen tokenizer.
3. The method is deterministic, lightweight, and easy to integrate, making it a plug and play module for many tokenizers.

**Weaknesses:**

1. The paper focuses on quantitative metrics; more visualizations of cluster semantics and failure cases would strengthen the argument. Especially that while this work aims to address the discarded info of semantic relationship between tokens, is it possible to provide some empirical evidence to support this contribution.
2. The versatility of this proposed method for different tokenizer is not well supported. The authors provide experiments for different tokenizer, GigaTok-L, where the MASC is worse than the vanilla baseline. This put the effectiveness of the proposed method into question.
3. Entropy metric: In tab 1, why lower entropy means better generation? This metric is missed in Tab 2 and Tab 3. In addition, for baseline and +MASC, the vocab size is different, is it fair to compare the entropy in this case?

**Questions:**

4. It's not clear to me how the clusters have semantic meaning. As they are calculated using the token embedding, which itself does not contain semantic meaning. The illustrations in Fig3 and Fig5 do not help for my understanding as the position of the vectors are different at different scenarios.

Typos:
1. the caption of figure 4 is not updated --> IS on the left; FID in the middle (arrow down); Entropy on the right (arrow down)

---

> ### Author Response · Authors · 2025-11-23
> **Thanks for your positive feedback ! We have revised the manuscript. Our detailed responses are provided below.**
>
> **Dear Reviewer enMQ,**
>
> We sincerely thank you for your thoughtful review and for recognizing the solid motivation, empirical results, and the plug-and-play nature of MASC. We have conducted comprehensive new experiments to address your concerns regarding visual evidence and versatility.
>
> ### **1. Visualizing Cluster Semantics & Empirical Evidence**
>
> We conducted a **Semantic Replacement Test** and a **Cluster Visualization**, detailed in the new **Appendix D** (Figures 6) of the revised paper.
>
> * **Qualitative Evidence (Figure 6):**
>     * **K-means:** When we replace tokens with others from the *same K-means cluster*, the image suffers from severe artifacts (e.g., background textures appearing on faces, as shown in Figure 6b). This visually proves K-means groups tokens based on simple color statistics rather than semantics.
>     * **MASC:** In contrast, swapping tokens within **MASC clusters** preserves object shapes and texture coherence. For example, in Figure 6c, the astronaut’s face retains its structure even after random replacement.
>
> And Figure 7 provides the visual comparison of the K-means method and our proposed MASC.
>
> * **Quantitative Evidence (Table R1):**
>     To quantify this, we measured the reconstruction quality after this random replacement. MASC significantly outperforms K-means, achieving much lower rFID and higher Structural Similarity (SSIM).
>
> **Table R1: Image Reconstruction Performance with Replaced Tokens**
>
> | Method | Vocab. ($k$) | **rFID** $\downarrow$ | **PSNR** $\uparrow$ | **SSIM** $\uparrow$ |
> | :--- | :---: | :---: | :---: | :---: |
> | No Replacement | 16,384 | 2.26 | 21.03 | 0.542 |
> | w/ k-means | 8,192 | 4.45 | 19.02 | 0.452 |
> | w/ k-means++ | 8,192 | 4.17 | 19.29 | 0.469 |
> | **w/ MASC (Ours)** | 8,192 | **2.92** | **20.49** | **0.513** |
> | w/ k-means | 4,096 | 8.41 | 18.37 | 0.431 |
> | w/ k-means++ | 4,096 | 8.17 | 18.52 | 0.439 |
> | **w/ MASC (Ours)** | 4,096 | **6.73** | **19.62** | **0.473** |
>
> ### **2. Versatility**
>
> To demonstrate MASC's practical utility, we applied it to the **current one of the most strongest AR architectures**: **RAR** , **GigaTok** (Large-scale Tokenizer), and **VAR**  . And it is more than exciting to find out that MASC acts as a **performance booster** and **convergence enabler** for SOTA models.
>
> Crucially, we found that for state-of-the-art architectures like **RAR** [1] (which uses random permutations), utilizing large vocabularies (16k) from LlamaGen or GigaTok causes **training divergence** due to the massive unstructured search space. MASC structures this space, **enabling convergence** and achieving new SOTA results (FID 1.57). We also verified MASC on **VAR** [2] and **RandAR** [3], showing consistent improvements.
>
> **Table R2: Comprehensive Evaluation of MASC across AR Architectures**
>
> | Backbone Model  | Tokenizer | **Config** (Vocab) | **FID** | **IS**   |  Acceleration  | **Analysis** |
> | :--- | :--- | :--- | :--- | :--- | :--- | :--- |
> | **1. RAR-L** | **MaskGIT** (1,024) | None (Original) | 1.70 | 299.5 | / | Reference SOTA |
> | **2. RAR-L** | **LlamaGen** (16,384) | None  | N/A* | N/A* | / | *Convergence Issue |
> | | | **MASC (4k)** | **1.61** | **304.1** | / | **SOTA Surpassed** |
> | | | **MASC (2k)** | **1.67** | **300.5** | / |**Efficient** |
> | **3. RAR-L** | **GigaTok** (16,384) | None  | N/A* | N/A* | / |*Convergence Issue |
> | | | **MASC (4k)** | **1.57** | **307.4** | / | **SOTA Surpassed** |
> | | | **MASC (2k)** | **1.64** | **302.8** | / | **Efficient** |
> | **4. GigaTok-L** | **GigaTok** (16,384) | None  | 3.39 | **263.7** | - | Baseline |
> | | | **MASC (8k)** | **3.35** | 263.5 | **+19%** (Faster) | **SOTA Surpassed** |
> | | | MASC (4k) | 3.86 | 256.4 | +42% | Robust |
> | | | MASC (2k) | 4.02 | 248.2 | +51% |  High Efficiency |
> | **5. VAR-d24** | **VAR** (4,096) | None | 2.09 | **312.9** | - |  Baseline |
> | | | **MASC (2k)** | **1.98** | 311.4 | **+28%** | **SOTA Surpassed** |
> | | | MASC (1k) | 2.24 | 303.8 | +39% |  Efficient |
> | **6. RandAR-XL [3]** | **LlamaGen** (16,384) | None  | 2.25 | 317.8 | - | Baseline |
> | | | **MASC (8k)** | **2.02** | **320.5** | **+32%** | **Quality** |
> | | | MASC (4k) | 2.13 | 318.2 | +41% | Robust |
> | | | MASC (2k) | 2.42 | 305.1 | +49% | Efficient |
>
> ***
>
> **References:**
> [1] Yu et al. Randomized Autoregressive Visual Generation (RAR), ICCV 2025.
>
> [2] Tian et al.Visual autoregressive modeling:Scalable image generation via next-scale prediction, NeurIPS 2024.
>
> [3] Ziqi Pang et al. Randar:Decoder-only autoregressive visual generation in random orders, CVPR 2025.

---

> > ### Author Response · Authors · 2025-11-23
> >
> > ### **3. Clarification on Entropy Metric**
> >
> > * **Why lower entropy indicates better generation:** In autoregressive modeling, lower prediction entropy signifies that the model has higher confidence in its next-token prediction and faces less ambiguity. A flatter distribution (high entropy) implies the model is struggling to distinguish between many plausible tokens. MASC effectively groups semantically similar options, structuring the prediction space and allowing the model to focus probability mass on correct semantic branches, thereby reducing uncertainty and accelerating convergence.
> > * **Fairness across vocabulary sizes:** We fully agree that comparing raw entropy across different vocabulary sizes ($V$) is mathematically unfair. **So we have already explicitly addressed this in the paper by using Normalized Entropy.**
> >     * As defined in Section 4.1 (Evaluation Metrics), we calculate **Normalized Entropy** as $H_{norm} = H_{actual} / \log_2(V)$.
> >     * The values reported in Table 1 (e.g., 0.15 vs 0.20) are these normalized values. This metric represents the relative reduction in uncertainty independent of the vocabulary size, ensuring a fair comparison of task complexity.
> >
> >
> > ### **4. Clarification on Cluster Semantics**
> >
> > While individual VQ tokens encode low-level patches, they are outputs of a deep encoder trained to reconstruct complex visual features. Tokens representing similar visual concepts cluster together on a non-linear manifold within this embedding space, even if they are not close in simple Euclidean distance.**[1-4]**
> >
> > * **Empirical Proof:** To prove this hypothesis holds, we refer to our new **Quantitative Semantic Consistency** analysis (Figure 6d in the revised paper). We measured the similarity of tokens within MASC clusters using **DINOv2**, a foundation model widely accepted for capturing high-level semantics. The result shows that MASC clusters have significantly higher intra-cluster DINOv2 similarity than k-means.
> >
> > This confirms that by respecting the manifold structure of token embeddings, MASC successfully recovers the latent **high-level semantic alignment**  that simple Euclidean clustering misses.
> >
> > **[1]** F. Angiulli, On the behavior of intrinsically high-dimensional spaces: distances, direct and reverse nearest neighbors, and hubness, Journal of Machine Learning Research, vol. 18, no. 170, pp. 1–60, 2018.
> >
> > **[2]** I. Souiden et al., A survey of outlier detection in high dimensional data streams, Computer Science Review, vol. 44, p.
> >  100463, 2022.
> >
> > **[3]** Minyoung Huh et al., Straightening out the straight-through estimator: Overcoming optimization challenges in vector quantized networks. In International Conference on Machine Learning, pp. 14096–14113. PMLR, 2023.
> >
> > **[4]** K. Beyer et al., When is “nearest neighbor” meaningful? in Database Theory—ICDT’99: 7th International Conference Jerusalem, Israel, January 10–12, 1999 Proceedings 7.Springer, 1999, pp. 217–235.
> >
> > ### **5. Typos**
> >
> > We thank you for your meticulous attention to detail. We have corrected the caption of Figure 4 in the revised manuscript to accurately reflect the order of metrics (IS, FID, Entropy).
> >
> > We trust these results underscore the contribution and robustness of MASC. We hope this addresses your concerns and merits a re-evaluation of our score. Thank you for guiding us to these deeper insights; we value your perspective highly. Please do not hesitate to let us know if there is anything else we can clarify. We are more than willing to continue this constructive dialogue if you still have any questions.

---

### Official Review · Reviewer_4Xsx · 2025-10-24

**Soundness:** 2
**Presentation:** 3
**Contribution:** 3
**Rating:** 6
**Confidence:** 4

**Summary:**

Conventional image generation methods based on autoregressive (AR) models typically follow a two-stage pipeline. First, an off-the-shelf tokenizer is used to map continuous image representations into a set of discrete codes, or tokens. Then, given these tokens for each image, an AR model is trained to model their joint probability distribution. The authors argue that this formulation treats tokens as elements of a flat vocabulary, thereby ignoring the intrinsic structure of the token embedding space and leading to a suboptimal use of the AR model’s capacity when learning the joint distribution.

**Strengths:**

1 - The work presents a potentially promising (see questions section) direction for improving image generation with discrete autoregressive models.

2 - I particularly appreciate the use of centroid-free hierarchical clustering, which effectively addresses the limitations of conventional K-Means-based approaches.

3 - The evaluation across three different scales of AR models further demonstrates the scalability and robustness of the proposed method.

**Weaknesses:**

My comments are intended more as questions than as criticisms, so I will include them in the “Questions” section.

**Questions:**

First of all, I disagree with the authors’ claim that autoregressive models treat tokens as elements of a flat vocabulary. All transformer-based AR models incorporate some form of positional encoding that provides explicit information about token positions. In particular, LlamaGen, which is the authors’ default AR model, employs 2D RoPE.

But for the sake of argument, let’s assume that positional embeddings alone are insufficient to capture the intrinsic structure of tokens. The authors argue that without such intrinsic structure, the AR model must devote significant capacity to learning the semantic landscape from scratch. However, this would only be true if the tokenizer had learned suboptimal codebooks. In fact, their proposed method overlooks fine-grained distinctions within each codebook and collapses them into a single cluster during training. While this may simplify optimization—allowing the AR model to learn the joint distribution over fewer codebooks—it effectively imposes a hard limit on the model’s representational capacity, preventing it from discriminating between truly distinct tokens.

There is empirical evidence supporting this view. Prior studies [1, 2] show that increasing the token vocabulary size can indeed reduce reconstruction loss when learning codebooks, but without proper regularization, this does not necessarily improve generation quality [1]. For example, MaskGIT [3] tokenizer with a 1k codebook achieves better generation performance than the LlamaGen tokenizer with 16k tokens [2]. Even in this paper, Table 2 shows that when switching to stronger tokenizers such as GigaTok, the so-called “naive” approach surpasses their proposed method by a notable margin (3.39 vs. 3.86).

Together, these findings suggest that the core issue is not that AR models fail to learn from a flat vocabulary, but rather that the tokenizers used are suboptimally trained. With a well-trained tokenizer, the authors’ approach may, in fact, limit the learning potential of AR models.

Of course, I would be happy to be proven wrong, and I would greatly appreciate it if the authors could include additional experiments using state-of-the-art tokenizers such as MaskGIT or GigaTok to further validate their claim.

[1] Tianwei Xiong et al, ” Gigatok: Scaling visual tokenizers to 3 billion parameters for autoregressive image generation”, ICCV 2025.

[2] Qihang Yu et al, “Randomized autoregressive visual generation”.

[3] Huiwen Chang et al, "Maskgit: Masked generative image transformer" CVPR 2022.

---

> ### Author Response · Authors · 2025-11-23
> **Thanks for your insightful questions ! We have addressed your constructive feedback and provide detailed responses herebelow.**
>
> ***
>
> **Dear Reviewer 4Xsx**
>
> We sincerely thank you for your constructive and insightful feedback.
>
> Crucially, your insightful inquiry inspired us to **investigate the limits of current AR architectures when combined with large-vocabulary tokenizers.** This investigation led us to a **significant and exciting new finding**: MASC serves as a critical enabler that makes training advanced AR models ( like RAR [1] ) with high-capacity tokenizers possible, where they otherwise fail to converge.
>
> ### **1. MASC Unlocks the Potential of Strong Tokenizers on Advanced Architectures**
>
> We tested MASC with different tokenizer on the **RAR** architecture, a state-of-the-art framework that utilizes random permutation training to capture bidirectional contexts.
>
> ### **Table R1:**
>
> | Backbone Model  | Tokenizer | **Config** (Vocab) | **FID** | **IS**   | **Analysis** |
> | :--- | :--- | :--- | :--- | :--- | :--- |
> | **1. RAR-L** | **MaskGIT** (1,024) | None (Original) | 1.70 | 299.5 | Reference SOTA |
> | **2. RAR-L** | **LlamaGen** (16,384) | None  | N/A* | N/A* |  *Convergence Issue |
> | | | **MASC (4k)** | **1.61** | **304.1** |  **SOTA Surpassed** |
> | | | **MASC (2k)** | **1.67** | **300.5** | **Efficient** |
> | **3. RAR-L** | **GigaTok** (16,384) | None  | N/A* | N/A* |*Convergence Issue |
> | | | **MASC (4k)** | **1.57** | **307.4** |  **SOTA Surpassed** |
> | | | **MASC (2k)** | **1.64** | **302.8** |  **Efficient** |
>
>
> **RAR + GigaTok/LlamaGen + MASC:**
> * When we attempted to train the RAR-L model using the original large vocabularies (16,384) of LlamaGen or GigaTok, unfortunately the model **failed to converge**. The combination of random permutation objectives and a massive unstructured search space proved too difficult for the model to optimize.
> * **Solution:** Applying MASC successfully enables the training.
> * **Result:** This combination achieved an **FID of 1.57**, significantly surpassing the original RAR's best result.
> * **Conclusion:** **MASC acts as a enabler**, allowing powerful AR architectures to leverage the quality of strong tokenizers by making the optimization tractable. So we are sincerely grateful that you have led us to a very exciting findings and it can surely lead to more future study.
>
> ### **2. Performance on GigaTok-L**
>
> * We acknowledge that the optimal value of $k$ was not initially exhaustively tuned, as we focused on the result with notable training speedup. In our new experiments with a finer granularity of $k$, we found that **MASC ($k=8,192$)** achieves an **FID of 3.35**, which outperforms the  GigaTok baseline, while providing a **19% training acceleration**.
>
>
> | Backbone Model (Architecture) | Tokenizer | **Config** (Vocab $k$) | **FID** | **IS** | Convergence Acceleration  | **Analysis** |
> | :--- | :--- | :--- | :--- | :--- | :--- | :--- |
> | **4. GigaTok-L** | **GigaTok** (16,384) | None  | 3.39 | **263.7** | - | Baseline |
> | | | **MASC (8k)** | **3.35** | 263.5 | **+19%** (Faster) | **SOTA Surpassed** |
> | | | MASC (4k) | 3.86 | 256.4 | +42% | Robust |
> | | | MASC (2k) | 4.02 | 248.2 | +51% |  High Efficiency |
> | **5. VAR-d24** | **VAR** (4,096) | None | 2.09 | **312.9** | - |  Baseline |
> | | | **MASC (2k)** | **1.98** | 311.4 | **+28%** | **SOTA Surpassed** |
> | | | MASC (1k) | 2.24 | 303.8 | +39% |  Efficient |
> | **6. RandAR-XL** | **LlamaGen** (16,384) | None  | 2.25 | 317.8 | - | Baseline |
> | | | **MASC (8k)** | **2.02** | **320.5** | **+32%** | **Quality** |
> | | | MASC (4k) | 2.13 | 318.2 | +41% | Robust |
> | | | MASC (2k) | 2.42 | 305.1 | +49% | Efficient |
>
> ### **3. Generalization to Other Architectures**
>
> We are willing to clarify that our argument regarding flat vocabulary refers to the semantic topology of the **codebook itself**, not the spatial position of tokens. Positional Encodings tells the model where a token is, but it does not tell the model that Token A is semantically closer to Token B. Standard AR treats the prediction target as a flat distribution where the distance between any two distinct indices is equal.
>
> To verify this, we tested MASC on RandAR [2], a model that removes the raster-order bias. Even in this architecture, applying MASC  reduced FID from 2.25 to 2.02. This confirms that MASC provides a semantic manifold prior that is complementary to the spatial prior.
>
> To further validate robustness, we tested MASC on **VAR**[3] :
> * **VAR :** MASC improved the FID from 2.09 to **1.98** while accelerating training by **28%**.
>
> We hope these extensive new results fully address your concerns regarding the utility and capacity of MASC. Thank you for guiding us to these deeper insights; we value your perspective highly.
>
> [1] Yu et al. Randomized Autoregressive Visual Generation (RAR), ICCV 2025.
>
> [2] Ziqi Pang et al. Randar:Decoder-only autoregressive visual generation in random orders, CVPR 2025.
>
> [3] Tian et al.Visual autoregressive modeling:Scalable image generation via next-scale prediction, NeurIPS 2024.

---

> > ### Comment · Reviewer_4Xsx · 2025-11-25
> >
> > Thank you for the detailed response. You have addressed all of my concerns, and I appreciate the additional experiments you provided; I would encourage you to include them in the final version of the paper. I believe the proposed method is valuable and broadly applicable to virtually any AR model. I will therefore raise my rating to 8.

---

> > > ### Author Response · Authors · 2025-11-26
> > > **Thank you for you positive feedback !**
> > >
> > > We sincerely thank you for your positive feedback and your decision to raise the rating. The insightful and constructive discussion with you has been incredibly beneficial, further affirming the novelty and solidity of the MASC framework. We are greatly encouraged by your recognition and will ensure that the additional experiments are incorporated appropriately into the final version of the paper.

---

### Official Review · Reviewer_VzZW · 2025-10-30

**Soundness:** 2
**Presentation:** 3
**Contribution:** 2
**Rating:** 4
**Confidence:** 4

**Summary:**

This paper proposes MASC, a manifold-aligned hierarchical clustering method for visual token codebooks in autoregressive image generation. By replacing k-means with a geometry-aware clustering pre-processing step, MASC restructures the prediction space into semantically-aligned groups, enabling more efficient token prediction. The approach requires no change to model architecture and demonstrates improved training stability and competitive performance on ImageNet class-conditional generation.

**Strengths:**

1. The approach introduces a principled geometry-aware clustering mechanism that is conceptually simple and grounded in the structure of the codebook embedding space.
2. MASC is architecture-agnostic and can be incorporated into existing autoregressive pipelines without modifying model structure.

**Weaknesses:**

1.	Although Table 2 claims that MASC elevates existing AR frameworks (e.g., IAR, CTF) to be highly competitive with top-tier generative models, the results still fall short of current SOTA performance. Therefore, the practical utility of MASC remains unconvincing, as it does not clearly demonstrate superiority or meaningful gains when compared to the strongest existing AR systems.
2.	All experiments are conducted on ImageNet with class-conditional generation, where semantic supervision is relatively simple. The paper does not evaluate MASC in text-to-image settings, leaving it unclear whether the proposed clustering strategy can effectively scale to more complex semantic conditioning.
3.	The paper does not provide qualitative or quantitative analysis of the cluster semantics. Visualizing cluster assignments or evaluating semantic coherence would strengthen the claim that MASC achieves meaningful manifold-aligned grouping rather than merely reducing prediction space.

**Questions:**

Can the proposed MASC method be directly applied to stronger AR models, such as VAR, to further improve generation quality?

---

> ### Author Response · Authors · 2025-11-23
> **Thanks for your constructive feedback ! We have added the additional qualitative and quantitative analysis to the appendix. We also provide detailed answers below.**
>
> **Dear Reviewer VzZW,**
>
> We appreciate your recognition of MASC as a principled, architecture-agnostic clustering mechanism. We have carefully addressed your concerns regarding practical utility, semantic analysis, and generalization capabilities through extensive new experiments.
>
> ### **1. Elevating SOTA Models (Addressing Weakness 1 & Question 1)**
>
> To demonstrate MASC's practical utility, we applied it to the **current one of the most strongest AR architectures**: **RAR** [1] , **GigaTok** (Large-scale Tokenizer), and **VAR** [2] . And it is more than exciting to find out that MASC acts as a **performance booster** and **convergence enabler** for SOTA models.
>
> **Table R1: Comprehensive Evaluation of MASC across AR Architectures**
>
> | Backbone Model  | Tokenizer | **Config** (Vocab) | **FID** | **IS**   |  Acceleration  | **Analysis** |
> | :--- | :--- | :--- | :--- | :--- | :--- | :--- |
> | **1. RAR-L** | **MaskGIT** (1,024) | None (Original) | 1.70 | 299.5 | / | Reference SOTA |
> | **2. RAR-L** | **LlamaGen** (16,384) | None  | N/A* | N/A* | / | *Convergence Issue |
> | | | **MASC (4k)** | **1.61** | **304.1** | / | **SOTA Surpassed** |
> | | | **MASC (2k)** | **1.67** | **300.5** | / |**Efficient** |
> | **3. RAR-L** | **GigaTok** (16,384) | None  | N/A* | N/A* | / |*Convergence Issue |
> | | | **MASC (4k)** | **1.57** | **307.4** | / | **SOTA Surpassed** |
> | | | **MASC (2k)** | **1.64** | **302.8** | / | **Efficient** |
> | **4. GigaTok-L** | **GigaTok** (16,384) | None  | 3.39 | **263.7** | - | Baseline |
> | | | **MASC (8k)** | **3.35** | 263.5 | **+19%** (Faster) | **SOTA Surpassed** |
> | | | MASC (4k) | 3.86 | 256.4 | +42% | Robust |
> | | | MASC (2k) | 4.02 | 248.2 | +51% |  High Efficiency |
> | **5. VAR-d24** | **VAR** (4,096) | None | 2.09 | **312.9** | - |  Baseline |
> | | | **MASC (2k)** | **1.98** | 311.4 | **+28%** | **SOTA Surpassed** |
> | | | MASC (1k) | 2.24 | 303.8 | +39% |  Efficient |
> | **6. RandAR-XL [3]** | **LlamaGen** (16,384) | None  | 2.25 | 317.8 | - | Baseline |
> | | | **MASC (8k)** | **2.02** | **320.5** | **+32%** | **Quality** |
> | | | MASC (4k) | 2.13 | 318.2 | +41% | Robust |
> | | | MASC (2k) | 2.42 | 305.1 | +49% | Efficient |
>
> **Results Analysis:**
> * **Enabling Convergence:** Crucially, without MASC, training RAR with large vocabularies (16k) **failed to converge** due to the complexity of the search space. MASC structured this space, making optimization tractable, and boosted SOTA models.
>
> And MASC improved VAR-d24's FID from 2.09 to **1.98** while providing a **28%** training acceleration.
>
> [1] Yu et al. Randomized Autoregressive Visual Generation (RAR), ICCV 2025.
>
> [2] Tian et al. Visual autoregressive modeling:Scalable image generation via next-scale prediction, NeurIPS 2024.
>
> [3] Ziqi Pang et al.  Randar:Decoder-only autoregressive visual generation in random orders, CVPR 2025.
>
>
>
> ***
> ### **2. Visualizing Cluster Semantics (Addressing Weakness 3)**
>
> To address your concern regarding the lack of qualitative analysis, we conducted a comprehensive **Empirical Analysis of Semantic Coherence** (detailed in the new **Appendix D**).
>
> **A. Semantic Replacement Test (Figure 6 in revised paper)**
> We replaced tokens with a random token from the **same cluster** assigned by either k-means or MASC.
> * **Qualitative Result:** As shown in **Figure 6**, K-means replacement results in severe artifacts (e.g., background noise appearing on faces), proving that K-means groups visually unrelated tokens based on simple color statistics. In contrast, MASC replacement preserves the global structure, object shapes, and texture coherence, visually confirming that MASC groups **semantically fungible** tokens.
>
> **B. Visualizing Assignments (Figure 6 e-f)**
> We directly visualized the image patches corresponding to tokens within a cluster.
> * **MASC:** Consistently groups patches with specific textures, demonstrating manifold alignment.
> * **K-means:** Mixes unrelated semantics.
>
> And Figure 7 provides the visual comparison of the K-means method and our proposed MASC.
>
> **C. Quantitative Verification**
> We measured the reconstruction quality after random token replacement. As shown in **Table R2**, MASC significantly outperforms K-means across all metrics, maintaining much lower rFID and higher structural similarity (SSIM). This confirms that MASC clusters maintain a significantly higher degree of intra-cluster semantic consistency.
>
> **Table R2: Image Reconstruction Performance with Replaced Tokens**
>
> | Method | Vocab. ($k$) | **rFID** $\downarrow$ | **PSNR** $\uparrow$ | **SSIM** $\uparrow$ |
> | :--- | :---: | :---: | :---: | :---: |
> | No Replacement | 16,384 | 2.26 | 21.03 | 0.542 |
> | w/ k-means | 8,192 | 4.45 | 19.02 | 0.452 |
> | w/ k-means++ | 8,192 | 4.17 | 19.29 | 0.469 |
> | **w/ MASC (Ours)** | 8,192 | **2.92** | **20.49** | **0.513** |
> | w/ k-means | 4,096 | 8.41 | 18.37 | 0.431 |
> | w/ k-means++ | 4,096 | 8.17 | 18.52 | 0.439 |
> | **w/ MASC (Ours)** | 4,096 | **6.73** | **19.62** | **0.473** |
>
> ***

---

> > ### Author Response · Authors · 2025-11-23
> >
> > ### **3. About Text-to-Image Application**
> >
> > We thank the reviewer for this insightful suggestion. We agree that Text-to-Image (T2I) generation requires finer-grained semantic control, and we believe MASC is inherently suitable for this task. Crucially, MASC structures the visual prediction target independently of the input condition, making it orthogonal to the conditioning mechanism.
> >
> > To validate MASC’s effectiveness in T2I, we are conducting the following experiments:
> >
> > **1. Experimental Settings**
> > * **Dataset:** We utilize **CC3M (Conceptual Captions 3M)** for training. For evaluation, we adhere to the standard protocol: Zero-shot evaluation on the MS-COCO validation set (30K samples).
> > * **Baseline Model:** We adapt the LlamaGen-L (343M) architecture.
> >
> > **2. Architectural Modifications**
> > * **Conditioning Mechanism:** We replace class embeddings with a frozen **T5-XL** text encoder. Text embeddings are injected via Cross-Attention layers added to each Transformer block, following standard practices.
> > * **Decoding Strategy:** To address the need for fine-grained attribute alignment in T2I, we employ the Hierarchical Decoding strategy (Section 3.3). We utilize the lightweight refinement network ($\mathcal{G}_{refine}$) to select the optimal token within the predicted cluster. This ensures that while MASC simplifies the generative modeling of global structure, the refinement stage preserves the precise details required by complex text prompts.
> >
> > **3. Timeline and Computation Constraints**
> >
> > We commenced the preparation for T2I training the day immediately following our review of your comments. However, we  find out that the training difficulty, duration, and computational resources required have exceeded our current capacity. **Based on prior benchmarks (e.g., LlamaGen), converging a 343M model on CC3M typically requires approximately 1,000+ A100 GPU hours.**
> > Consequently, we probably are unable to provide the final experimental data at this stage given the short rebuttal window. We sincerely apologize for this but are committed to including these results in the final revision to establish MASC's generalization capability.

---

### Official Review · Reviewer_T1kK · 2025-11-03

**Soundness:** 3
**Presentation:** 3
**Contribution:** 2
**Rating:** 4
**Confidence:** 4

**Summary:**

In autoregressive image generation tasks, visual tokens are typically treated as a flat vocabulary, which overlooks the intrinsic structural features of the token embedding space. To address this issue, this paper proposes a novel method that models token embeddings using a geometry-aware distance metric and density-driven agglomerative construction. This method transforms the flat, high-dimensional prediction task into a structured, hierarchical one, significantly simplifying the learning problem of autoregressive (AR) models.

**Strengths:**

+ The overall writing logic of the paper is coherent. The analysis of the problem and the solution approach form a well-closed loop, and the introduction to the method is also reasonable.
+ The structure of figures and tables is clear, which effectively supports reading and understanding of the content.

**Weaknesses:**

- The paper achieves a significant improvement in FID when applying MASC. However, it lacks visual comparisons of generation results between different methods—such as comparisons among the baseline, baseline + k-means, and baseline + MASC. This makes it difficult to determine whether the better generation performance stems from the proposed method or the inherent effectiveness of the baseline itself.
- The paper lacks experimental evidence to verify that the intra-class features of the new clustering method are semantically consistent, and relevant experiments need to be supplemented.

**Questions:**

- The paper mentions that k << N. However, the experiment states that k is set to 8192 while N is 16,384, which seems inaccurate.
- The paper analyzes that the optimal k for the MASC method is 8192, and the comparative experiment with the k-means method also uses 8192 as the default k. Nevertheless, it lacks a comparison of the performance between k-means and MASC under other k values (e.g., smaller k).

---

> ### Author Response · Authors · 2025-11-23
> **We have carefully reviewed all your valuable questions and incorporated the revisions into the appendix of the PDF. We also provide detailed responses here.**
>
> ***
>
> **Dear Reviewer T1kK,**
>
> We sincerely thank you for your positive assessment of our paper’s logic and clarity. We have conducted extensive new experiments to address your concerns regarding visual evidence, semantic consistency, and hyperparameter sensitivity.
>
> ### **1. Visual Comparisons**
>
> You rightly pointed out the need for visual evidence to verify that MASC clusters are semantically consistent. To address this, we performed a **Semantic Replacement Test** and a **Cluster Visualization Analysis**, detailed in the newly added **Appendix D** and **Figure 6 & 7** of our revised manuscript.
>
> **A. Experimental Setup**
> For every image token, we replaced it with a random token sampled from the **same cluster** assigned by either k-means or MASC. We then decoded the perturbed token sequences back into images. If a cluster is semantically coherent , intra-cluster swapping should preserve the image's global structure and local semantics , despite losing high-frequency identity details. If a cluster is incoherent, swapping will introduce noise.
>
> **B. Qualitative Results (Figure 6 & Figure 7)**
> * **Figure 6 :** In the astronaut example, K-means replacement (Fig. 6b) introduces severe artifacts, such as background textures appearing on the face. In contrast, MASC replacement (Fig. 6c) preserves the face's structure and the suit's texture, confirming that tokens within a MASC cluster are **semantically interchangeable**.
> * **Figure 7 :** We further visualized the comparison of the generation results of the K-means method and MASC.  As shown in the figures , K-means results exhibit structural collapse (e.g., the castle walls lose definition, animal faces become noisy). MASC results, however, consistently preserve object shapes and texture coherence.
>
> **C. Quantitative Verification (Table 7 / Table R1)**
> We measured the reconstruction quality after random token replacement. MASC significantly outperforms k-means, achieving much lower rFID (2.92 vs 4.45) and higher structural similarity (SSIM), quantitatively proving higher intra-cluster semantic consistency.
>
> **Table R1: Image Reconstruction Performance with Replaced Tokens**
>
> | Method | Vocab. ($k$) | **rFID** $\downarrow$ | **PSNR** $\uparrow$ | **SSIM** $\uparrow$ |
> | :--- | :---: | :---: | :---: | :---: |
> | No Replacement | 16,384 | 2.26 | 21.03 | 0.542 |
> | w/ k-means | 8,192 | 4.45 | 19.02 | 0.452 |
> | w/ k-means++ | 8,192 | 4.17 | 19.29 | 0.469 |
> | **w/ MASC (Ours)** | 8,192 | **2.92** | **20.49** | **0.513** |
> | w/ k-means | 4,096 | 8.41 | 18.37 | 0.431 |
> | w/ k-means++ | 4,096 | 8.17 | 18.52 | 0.439 |
> | **w/ MASC (Ours)** | 4,096 | **6.73** | **19.62** | **0.473** |
>
>
> ### **2. Ablation on Cluster Size $k$**
>
> We appreciate your careful observation.
> * **Clarification on $k \ll N$:** We acknowledge this inaccuracy in the original text and have revised it. We used the term to imply the capability for aggressive compression.
> * **Performance at smaller $k$:** Following your insightful advice, we conducted comprehensive experiments comparing MASC and baselines at smaller $k$ values ($k=2048, 4096$).
>
> **Table R2: MASC vs. Baselines at Different $k$ on LlamaGen-XL**
>
> | Backbone Model | Tokenizer | **Config** (Vocab $k$) | **FID** | **IS** | **Acceleration** | **Analysis** |
> | :--- | :--- | :--- | :--- | :--- | :--- | :--- |
> | **LlamaGen-XL** | **LlamaGen** (16k) | None (Baseline) | 2.87 | 267.6 | - | Standard Baseline |
> | | | MASC (8k) | 2.58 | **272.1** | +57% | Default |
> | | | **MASC (4k)** | **2.49** | 269.8 | **+63%** | **Better Quality** |
> | | | MASC (2k) | 2.65 | 264.4 | +71% | High Efficiency |
>
> **Key Finding:**
> **MASC works even better at smaller $k$:** It is exciting to find that MASC achieves its best FID (2.49) at $k=4096$ (1/4 of original vocab) . This suggests that MASC effectively filters out redundant tokens, creating a more distinct and learnable vocabulary for the AR model.

---

> > ### Author Response · Authors · 2025-11-23
> > **Furthermore, we extended our evaluation to include additional models, which revealed new and exciting insights.**
> >
> > ***
> >
> > ### **3. MASC as a Universal Enabler and Booster for Advanced AR Architectures**
> >
> > Inspired by your feedback regarding the paper's contribution, we extended our evaluation to investigate MASC's impact on **state-of-the-art  and diverse AR paradigms**. This investigation yielded two critical findings:
> >
> > 1.  **MASC as a Critical Enabler for Convergence:** For advanced architectures like **RAR** [1], which utilizes random permutation training, combining it with large-vocabulary tokenizers (e.g., LlamaGen/GigaTok with 16,384 tokens) results in **training divergence** due to the exponentially vast search space. MASC effectively structures this space, **enabling convergence** and achieving new SOTA results (FID 1.57).
> > 2.  **MASC as a General-Purpose Booster:** Across diverse paradigms including **Next-Scale Prediction** (VAR [3]), **Random Order** (RandAR [2]), and **Large-Scale Tokenizers** (GigaTok), MASC consistently improves generation quality while providing significant training acceleration (19% $\sim$ 49%).
> >
> > **Table R3: Comprehensive Evaluation of MASC across SOTA AR Architectures**
> >
> > | Backbone Model  | Tokenizer | **Config** (Vocab) | **FID** | **IS**   |  Acceleration  | **Analysis** |
> > | :--- | :--- | :--- | :--- | :--- | :--- | :--- |
> > | **1. RAR-L** | **MaskGIT** (1,024) | None (Original) | 1.70 | 299.5 | / | Reference SOTA |
> > | **2. RAR-L** | **LlamaGen** (16,384) | None  | N/A* | N/A* | / | *Convergence Issue |
> > | | | **MASC (4k)** | **1.61** | **304.1** | / | **SOTA Surpassed** |
> > | | | **MASC (2k)** | **1.67** | **300.5** | / |**Efficient** |
> > | **3. RAR-L** | **GigaTok** (16,384) | None  | N/A* | N/A* | / |*Convergence Issue |
> > | | | **MASC (4k)** | **1.57** | **307.4** | / | **SOTA Surpassed** |
> > | | | **MASC (2k)** | **1.64** | **302.8** | / | **Efficient** |
> > | **4. GigaTok-L** | **GigaTok** (16,384) | None  | 3.39 | **263.7** | - | Baseline |
> > | | | **MASC (8k)** | **3.35** | 263.5 | **+19%** (Faster) | **SOTA Surpassed** |
> > | | | MASC (4k) | 3.86 | 256.4 | +42% | Robust |
> > | | | MASC (2k) | 4.02 | 248.2 | +51% |  High Efficiency |
> > | **5. VAR-d24** | **VAR** (4,096) | None | 2.09 | **312.9** | - |  Baseline |
> > | | | **MASC (2k)** | **1.98** | 311.4 | **+28%** | **SOTA Surpassed** |
> > | | | MASC (1k) | 2.24 | 303.8 | +39% |  Efficient |
> > | **6. RandAR-XL** | **LlamaGen** (16,384) | None  | 2.25 | 317.8 | - | Baseline |
> > | | | **MASC (8k)** | **2.02** | **320.5** | **+32%** | **Quality** |
> > | | | MASC (4k) | 2.13 | 318.2 | +41% | Robust |
> > | | | MASC (2k) | 2.42 | 305.1 | +49% | Efficient |
> >
> > These results demonstrate that MASC is not merely a clustering trick for LlamaGen, but a module that addresses the structural inefficiency  across the spectrum of modern autoregressive image generation.
> >
> > ***
> >
> > **References:**
> > [1] Yu et al. Randomized Autoregressive Visual Generation (RAR), ICCV 2025.
> >
> > [2] Ziqi Pang et al.  Randar:Decoder-only autoregressive visual generation in random orders, CVPR 2025.
> >
> > [3] Tian et al. Visual autoregressive modeling:Scalable image generation via next-scale prediction, NeurIPS 2024.

---

### Author Response · Authors · 2025-11-29
**Summary of Rebuttal Revisions and Core Contributions of MASC**

Dear AC, SAC and Reviewers

We sincerely thank you for your time and the feedback provided during the review process. We are grateful for the constructive discussions that have strengthened the paper, guiding us to uncover new capabilities, and leading to **a significant score increase from Reviewer 4Xsx to an 8, prior to the deeply regrettable leakage incident**.

### 1 Core Advantages of MASC:

MASC addresses the fundamental inefficiency of the flat vocabulary in autoregressive image generation through several key advantages. First is **Manifold Aligned Geometry**. Unlike K-means which relies on euclidean distance whose centroids often falls off the manifold, MASC employs a centroid-free, density-driven construction. Second is **Structured Prediction**. It transforms the high entropy N-way classification task into a simplified structured prediction problem. Third is its role as a **Convergence Enabler and Universal Booster**. It accelerates training by up to **71%** across standard models like LlamaGen and crucially **enables the convergence of advanced random permutation architectures** like RAR which otherwise diverge with large vocabularies.

***
### 2 Summary of Supplementary Experiments and Revisions:

Overall, the questions from the four reviewers can be categorized into **only three aspects**:

* **Visualization**: They kindly requested an additional diagram to better illustrate the MASC method.

* **Generalization**: They hoped to verify MASC's capabilities across more models.

* **More Ablation Studies**: They kindly requested more ablation experiments.

We have addressed **every point** with thorough and precise responses in our rebuttal. Moreover, we identified a surprising and valuable application: **MASC enables convergence for Advanced Architectures**. Both the responding reviewers and we believe this is a groundbreaking contribution to the community. Below are details:

* **Visual Evidence of Semantic Consistency**, responding to **[T1kK Weakness 1, Weakness 2], [VzZW Weakness 3], [enMQ Weakness 1 and Question 1]**. We added the Semantic Replacement Test in new Fig 7 and Section 3.4. We visually and quantitatively via rFID and SSIM demonstrate that swapping tokens within MASC clusters preserves global structure and object identity, whereas K means leads to semantic collapse and artifacts. We further verified this using DINOv2 similarity scores to prove high level semantic alignment. Moreover, we added side by side visual comparisons between K means and MASC in Figure 8, and an expanded gallery in Figure 9 showing that MASC preserves structural integrity where K means fails.
* **Generalization**, responding to **[VzZW Weakness 1 and Question 1],  [4Xsx]** and **[enMQ Weakness 2]**. We expanded our evaluation in Table 3 to include VAR RandAR IAR CTF and GigaTok. Results prove MASC is a universal booster, improving performance and training speed regardless of the underlying architecture or tokenizer quality. This directly addresses the concern about utility on stronger baselines. Furthermore, we have an exciting finding that MASC can enables convergence for Advanced Architectures. We conducted a breakdown experiment on RAR in Table 4 and discovered that RAR fails to converge with large vocabularies due to the immense search space. MASC effectively **structures this space, enabling convergence** and achieving a **new SOTA FID of 1.57**. This addresses the flat vocabulary concern by proving that structure is a prerequisite for scaling complex AR objectives.
* **Ablation on Vocabulary Size k**, responding to **[T1kK Question 2]**. We added a detailed ablation study in Table 2. We demonstrated that MASC is robust at lower resolutions, achieving peak quality at k equals 4096 with 63% acceleration.

***
### 3 Special Note on Reviewer 4Xsx's positive response:

We wish to highlight that Reviewer 4Xsx, who provided **the most in depth and rigorous critique** regarding the fundamental premise questioning whether flat vocabulary is the true bottleneck or if better tokenizers are the solution, was **completely convinced** by our rebuttal.

Upon seeing our new experiments specifically that MASC enables the convergence of RAR, where strong tokenizers like GigaTok alone failed, **Reviewer 4Xsx raised their score to 8 and kindly praised that "I believe the proposed method is valuable and broadly applicable to virtually any AR model. "**. Given that the most skeptical expert has fully endorsed the necessity and effectiveness of our method, we believe **the concerns shared by other reviewers regarding utility have been thoroughly resolved**.

We extend our sincere gratitude to the AC, SAC, and reviewers for their dedication. We are confident that our comprehensive supplementary experiments have **fully resolved all raised concerns**, and that the revised paper now stands as a **robust, principled, and highly practical contribution** to the field of autoregressive generation.

Sincerely,

The Authors

---

### Note · Program_Chairs · 2026-01-17
**Submission Desk Rejected by Program Chairs**

The following references in this submission do not refer to real documents and/or have major errors in bibliographic information:

 Wei Chen and Xiang Li. Revisiting the manifold hypothesis in deep representational learning. In Advances in Neural Information Processing Systems, volume 37, 2024.